# India's agroecology programme, 'Zero Budget Natural Farming', delivers biodiversity and economic benefits without lowering yields

Iris Berger [1,2] ✉, Ajit Kamble[3], Oscar Morton [2,4,5], Varsha Raj[3], Sayuj R. Nair[3], David P. Edwards [2,4], Hannah S. Wauchope [6], Viral Joshi[7], Parthiba Basu [8,10], Barbara Smith [9] & Lynn V. Dicks [1,2]

The Global Biodiversity Framework promotes agroecological farming approaches[1], yet rigorous system-wide evaluations of agroecological programmes are urgently needed to balance the intertwined but partially competing Sustainable Development Goals of curbing food insecurity, improving human well-being and tackling biodiversity loss. Here we focus on the largest agroecological transition globally—the 64,000 km² government-incentivized zero budget natural farming (ZBNF) programme in India—to co-analyse socio-economic and biodiversity impacts. ZBNF more than doubled farmers' economic profits and maintained comparable crop yields. Bird biodiversity outcomes were improved, with the densities of bird species and functional guilds involved in pest control and seed dispersal increasing; however, natural forests remain essential to sustaining populations of forest-specialized species. Trade-offs between bird densities and landscape-scale yields and profit were substantially less pronounced in ZBNF than in conventional, agrichemical-based farming systems, underscoring the benefits of agroecological interventions with aligned protection of natural ecosystems.

Food systems world-wide are failing on a plethora of social, economic and environmental dimensions[2,3], with agriculture being the biggest driver of biodiversity loss[4,5]. Curbing future food demand is critical, for example, by transitioning to plant-based diets and reducing food waste[6–8].

Nonetheless, feeding the human population while limiting further conversion of natural habitats also necessitates high per-area agricultural productivity[9,10]. Equally, agricultural landscapes must urgently be redesigned and governed for social and environmental sustainability[11,12]. However, trade-offs between agricultural productivity and farmland biodiversity are prevalent[13,14] (although not inevitable[15–17]). Developing agricultural land systems that are simultaneously productive and environmentally sustainable is perhaps the greatest challenge of the twenty-first century[14,18].

There is growing recognition that commonly used means of increasing crop yield, such as agrichemicals and mechanization, are

[1]Department of Zoology, University of Cambridge, Cambridge, UK. [2]Conservation Research Institute, University of Cambridge, Cambridge, UK. [3]Independent Researcher, Mumbai, India. [4]Department of Plant Sciences and Centre for Global Wood Security, University of Cambridge, Cambridge, UK. [5]School of Biosciences, University of Sheffield, Sheffield, UK. [6]Global Change Institute, School of Geosciences, University of Edinburgh, Edinburgh, UK. [7]Indian Institute of Science Education and Research Tirupati, Tirupati, Andhra Pradesh, India. [8]Centre for Agroecology and Pollination Studies, Department of Zoology, University of Calcutta, Kolkata, West Bengal, India. [9]Centre for Agroecology, Water and Resilience, Coventry University, Coventry, UK. [10]Deceased: Parthiba Basu. ✉e-mail: irisberger1996@gmail.com

harming biodiversity and eroding ecosystem services which ironically trigger food productivity declines and decrease food security[19–21]. As a result, there is substantial interest in agroecological approaches that increase yield through the improvement of ecosystem services, both to meet sustainability goals but also to improve crop productivity. Such approaches could be pivotal nature-based solutions that simultaneously address food insecurity, social injustices, and the climate and biodiversity crises[22,23].

Most existing evaluations of agroecological approaches have either focused only on productivity impacts, been based on controlled field trials, or have not controlled for confounding variables that influence the likelihood of farmers adopting the approaches in the first place[24,25]. To assess whether these approaches constitute sustainable food solutions, a broader set of outcomes needs to be examined, including measures of biodiversity and profit. Importantly, outcomes also need to be measured in real-world, large-scale programmes and evaluated using robust causal inference methods to assess how policy decisions translate to impact on the ground.

Here we used social and biodiversity surveys to elucidate the impact of the world's largest agroecological transition[26–28], the zero budget natural farming (ZBNF) programme in Andhra Pradesh, South India. ZBNF aims to boost crop yields and reduce costs by ending the use of synthetic inputs and by regenerating the biotic interactions that underpin yield-supporting ecosystem services. Specifically, it involves four 'wheels': (1) 'jiwamrita' – the use of a microbial inoculum to stimulate microbial activity to make nutrients available to plants and protect against pathogens; (2) 'beejamrita' – the application of another microbial culture to protect young roots from fungal and soil-borne diseases; (3) 'acchadana' – mulching to produce stabilized soil organic matter, conserve top soil and increase the activity of soil biota; and (4) 'whapahasa' – increasing soil aeration by improving soil structure, and reduced overreliance on irrigation and tillage[27–29]. In addition, farmers are encouraged to maintain cover crops, diversify crop plants, intercrop the main crop with (herbaceous) plants that are repellent to pests, install bird perches, and plant hedgerows and farmland trees[28] (see Supplementary Information 1). State-wide conversion from conventional, agrichemical-based farming to ZBNF of Andhra Pradesh's 6 million farmer households (spread across ~64,000 km²) has been incentivized by the state government since 2016, primarily through training programmes[26,30]. With African and Latin American countries considering adopting this approach[31], there is an urgent need to assess the system's efficacy.

Our core objectives were thus to: (1) determine the ZBNF programme's impacts on crop yield and economic profit; (2) examine the ZBNF programme's effects on bird abundances, at trophic guild and species level; (3) assess whether trade-offs exist between yield/profit and bird abundances, and whether these trade-offs are dampened, or even neutralized, in ZBNF compared to agrichemical systems; and (4) to quantify how the bird communities in ZBNF and agrichemical systems each compare to those in natural forests. We studied wild birds as they generally reflect the biodiversity and ecological consequences of land-use (intensity) changes well[32,33], and most evaluations of density–productivity relationships have focused on them[14]. In brief, we visited 13 ZBNF, 13 agrichemical and 26 forest landscapes in northern Andhra Pradesh between 2021 and 2023, collecting data on crop yield and profit using social surveys, and bird abundances via point-count-based distance sampling. We matched ZBNF and agrichemical sampling units according to a range of covariates to eliminate rival explanations for our results and used seven different models to meet our four overarching objectives (see Methods and Supplementary Fig. 4 for a summary).

## Results and Discussion
### ZBNF does not affect yield, but boosts profit
Existing reports of yield benefits from ZBNF are based either on field plot experiments (for example, ref. 34), which cannot quantify

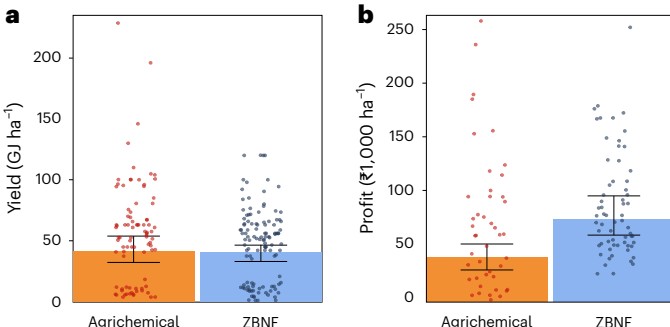

**Fig. 1 | Estimated harvest-level yield and field-level profit outcomes for agrichemical farming and ZBNF.** Bar charts depict weighted means for agrichemical farming (orange) and ZBNF (blue), whiskers represent the bias-corrected accelerated (BCa) bootstrap 95% CIs, and the scattered points represent the raw data. **a**, Yield did not differ significantly between the two farming practices ($n_{harvests}$ = 206, $n_{fields}$ = 128, $n_{landscapes}$ = 26), while **b**, profit was higher under ZBNF ($n_{fields}$ = 128, $n_{landscapes}$ = 26). Here, 'significant' refers to the 95% CIs of the estimated effect size not overlapping with zero.

system-wide socio-ecological effects or policy impact, or on observational studies that do not report efforts to control for confounding factors (for example, ref. 26). We controlled for observable confounders by careful site selection in the field and statistical matching, where we matched ZBNF to agrichemical farming sampling units, and re-ran the analysis multiple times to assess the sensitivity of our results to different parameter decisions (see Methods and Supplementary Information 6). Unobservable confounders were unlikely to substantially affect our conclusions (see Supplementary Information 5 for a justification). We combined data on productivity and costs for 206 harvests of 128 fields (situated within the 26 agricultural landscapes) and used g-computation (which entails comparing observed to counterfactual outcomes) to quantify the effect of the ZBNF programme on harvest-level crop yield and field-level economic profit (the former being a measure of each harvest and the latter encompassing the revenue and expenditure for all harvests grown on a given field annually).

On average, ZBNF did not reduce yield in comparison to agrichemical farming (estimated yield difference +1.5%, 95% confidence interval (CI): −5.6 to +19.3%) (Fig. 1a). This finding is robust to the matching specifications employed and whether the counterfactual system, that is, the type of farm that ZBNF was compared to, constituted lower-agrichemical-input subsistence farming or high-agrichemical-input farming (estimated yield difference −11.4% (−22.4 to +4.9%) and 1.20% (−3.6 to +6.6%), respectively; Supplementary Table 4). However, yields reported by ZBNF and agrichemical farmers varied markedly for both systems (Fig. 1a), suggesting productivity losses or benefits under certain scenarios. Care must be taken when rolling out the ZBNF programme in already high-yielding regions, and we stress the importance of future studies identifying the socio-economic and biophysical conditions and specific management interventions enabling ZBNF-induced yield benefits.

By lowering input costs, ZBNF increased economic profit by an average of 123.6% relative to agrichemical farming (95% CI: +63.1% to +244.0%; Fig. 1b), with a positive impact estimated under all versions of the analysis (Supplementary Table 5). A central aim of the ZBNF programme is to alleviate agrarian poverty[27], and these results suggest that the programme does so, although with considerable variation in the magnitude of the economic benefit received. Profitability boosts were delivered despite agrichemical subsidies and ZBNF produce not fetching market premiums, as reported by the farmers in our study (see Supplementary Information 3) and reflecting the state-wide situation[35]. A more enabling policy environment that eradicates agrichemical subsidies and creates ZBNF-specific value chains

would probably bring additional livelihood benefits[36]. Indeed, other social benefits of ZBNF have been noted, including improved food sovereignty, human health and gender equality[37,38], which are central pillars of sustainable food production systems, in particular for marginalized smallholder-dominated systems in the tropics.

We assessed the average impact of the ZBNF programme across the heterogeneity of practices covered by the programme and across socio-ecological settings, but our design did not enable us to ascertain the effect of individual ZBNF interventions on yield or profit. The intensity and frequency of employment of the four ZBNF 'wheels' spanned a continuum, with the types and quantities of ZBNF inputs tailored to each village (as recommended by the programme[27]). While the ZBNF programme encourages crop diversification, we found that ZBNF and agrichemical farmers grew the same crops (Supplementary Fig. 1) and the same number of crop types (likelihood ratio test (LRT), $P = 1$), and landscape-level crop diversity did not differ between the two agricultural systems (LRT, $P = 0.178$). Cover crops were used by around half (51.5%) of the ZBNF farmers, and a similar number (49.6%) installed bird perches. In contrast, few ZBNF farmers (7.4%) planted marigold to deter insect pests and none of them planted any hedgerows or farmland trees. Agrichemical farmers conducted none of these activities.

Finally, while agroecological practices frequently reduce interannual yield fluctuations[39–41], assessments of whether this resilience benefit occurs in ZBNF systems are urgently needed, especially since agriculture in the region is severely affected by climate change[42].

## ZBNF benefits farmland birds

To examine ZBNF's impact on bird abundances, we conducted 104 point counts (each with 4 repeats) situated in the 26 25-hectare landscapes, half (13) of which were managed under ZBNF and the other half (13) under agrichemical farming, with the respective farming system also dominating the wider area around each site (see Methods). Following statistical matching (as described above), we used a hierarchical Bayesian model to estimate the effect of the ZBNF programme on bird abundances at the trophic guild and species levels. We examined guild-level estimates to gain insights into community shifts and potential cascading ecological impacts, as previous research has found that agricultural intensification can markedly alter trophic structures and associated ecosystem functions and services[43,44], with species' functional traits influencing their persistence in agricultural landscapes[45,46]. We estimated species-level abundances to understand how ZBNF affected species of different habitat preferences and conservation importance (see Methods). In total, we recorded 114 bird species in agricultural landscapes, 48 of which were not recorded in forests (see below) and, of the 48 species, 13 were unique to ZBNF systems and 4 to agrichemical systems (Supplementary Table 6).

We show that, on average, ZBNF increased the abundance of frugivores by 160.25%, vertivores by 80.85%, and invertivores by 48.98% in comparison to agrichemical systems (Fig. 2a and Supplementary Table 9). On average, abundances of granivores and omnivores (comprising 13 and 27 species, respectively), which are generally more disturbance-resilient guilds[47,48], did not differ between ZBNF and agrichemical systems. Frugivorous birds are important seed dispersers

that facilitate the restoration of native and semi-native vegetation in agricultural landscapes[49], and could thus help reinstate landscape complexity, which has been declining across India[50]. The 8 frugivore species we recorded in agricultural systems, comprising 2 bulbuls, 2 barbets, an oriole, a hornbill, a parrot and a koel species (Supplementary Table 6), also feed on insects, small vertebrates and/or flower parts. Since ZBNF farmers did not grow more fruit-bearing woody or herbaceous vegetation, we speculate that the effect of the ZBNF programme on these species is more likely exerted via increases in these other dietary components. The lack of agrichemical-based pesticide, fertilizer and herbicide use in ZBNF systems may have benefited invertivorous and vertivorous birds via increased prey availability and, in turn, enhanced their ability to provide pest-control services that benefit food production and economic value[51,52]. Invertivores encompassed a diverse set of 41 species, including, for example, drongos, pipits, bee-eaters and warblers, whereas vertivorous birds comprised 24 primarily aquatic-affiliated species, such as kingfishers, herons and egrets (Supplementary Table 6).

At the individual-species level, 17 species were significantly more abundant in ZBNF than in agrichemical systems, while 6 were more abundant in agrichemical systems (where 'significant' means that at least 95% of the posterior share the same directional response as the median; Fig. 2b). The majority of species found more frequently in ZBNF systems were invertivores or vertivores; however, no single trait unified them all, with species varying in their dispersal ability, foraging and nesting behaviour, and life history (Extended Data Figs. 1–4 and Supplementary Table 6). Species included, for example, *Merops orientalis* (Asian Green Bee-eater), *Pycnonotus luteolus* (white-browed bulbul) and *Ardea alba* (great egret). Indeed, neither primary lifestyle (ground dwelling, perching or generalist) nor body mass drove responses to the ZBNF programme (Extended Data Figs. 1 and 2). Most of the species found more frequently in agrichemical systems were omnivores and granivores but, again, functionally diverse, including, for example, *Cinnyris asiaticus* (purple sunbird), *Columba livia* (rock dove) and *Euodice malabarica* (Indian silverbill) (Supplementary Table 8). These generalist species may have been more abundant in agrichemical systems due to heightened interspecies competition in ZBNF landscapes. The remaining 91 species had a non-significant response to ZBNF, although more species showed a tendency towards higher abundance in ZBNF (Fig. 2b and Supplementary Table 8). Farmland birds have sharply declined in India[53] and elsewhere[44,54], and these findings suggest that large-scale transitions to ZBNF might be an effective strategy to reverse some of these declines.

We recorded more species of conservation importance ('Near threatened' or 'Vulnerable' on the IUCN Red List and/or of national conservation priority[53]) in ZBNF compared with agrichemical systems (21 and 13, respectively; 22 combined). Sixteen (72.7%) of these 22 species were estimated to be more abundant, although non-significantly, in ZBNF systems (Fig. 2b). Most of these species (20; 90.9%) were primarily associated with non-forest habitats, including shrublands, grasslands and wetlands (Supplementary Table 6). Given that grasslands and wetlands are some of the most threatened ecosystems and are underrepresented in the conservation policies of India[55,56] and elsewhere[57,58],

**Fig. 2 | Effect of ZBNF on bird abundances at the trophic-guild level and species level.** The responses are expressed as a percentage change in abundance compared to agrichemical farming, where the median estimates and their 95% Bayesian credible intervals are shown. **a**, The abundance of the average frugivore, vertivore and invertivore was higher in ZBNF, whereas the abundance of the average granivore and omnivore did not differ. **b**, A total of 17 species increased in abundance, 6 decreased, and 91 species did not change. Species of conservation importance (IUCN Red List and/or species of national conservation priority) are highlighted. The colour of the error bars depicts a species' primary habitat (obtained from AVONET): forest (tall tree-dominated vegetation with more or less closed canopy), woodland (medium-stature tree-dominated habitats),

shrubland (low-stature bushy habitats), grassland (open dry to moist grass-dominated landscapes), wetland (wide range of freshwater aquatic habitats including lakes, marshes, swamps and reedbeds), riverine (associated with rivers and streams), rock (rocky substrate typically with no or very little vegetation) and human modified (urban landscapes, intensive agriculture, gardens). Error bars' opacity reflects the proportion of the posterior distribution (PD) that shares the same direction of response as the median (that is, confidence in the response), where 100% opacity reflects PD > 95%, and 50% opacity reflects no significant change. Species names are given in Supplementary Table 8. $n_{species} = 114$, $n_{point counts} = 416$, $n_{landscapes} = 26$.

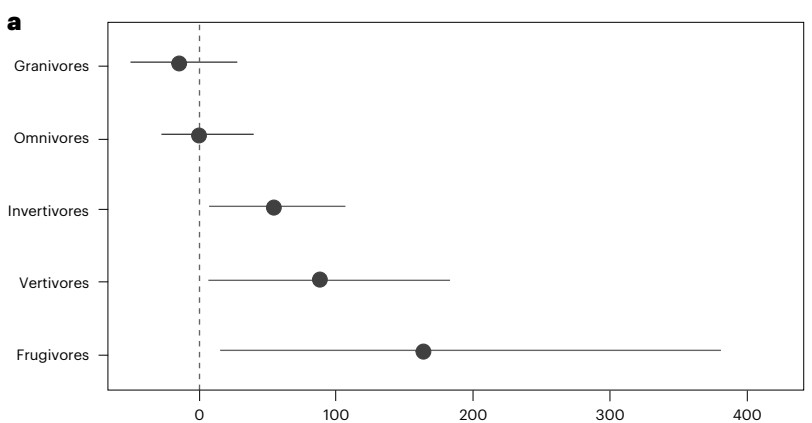

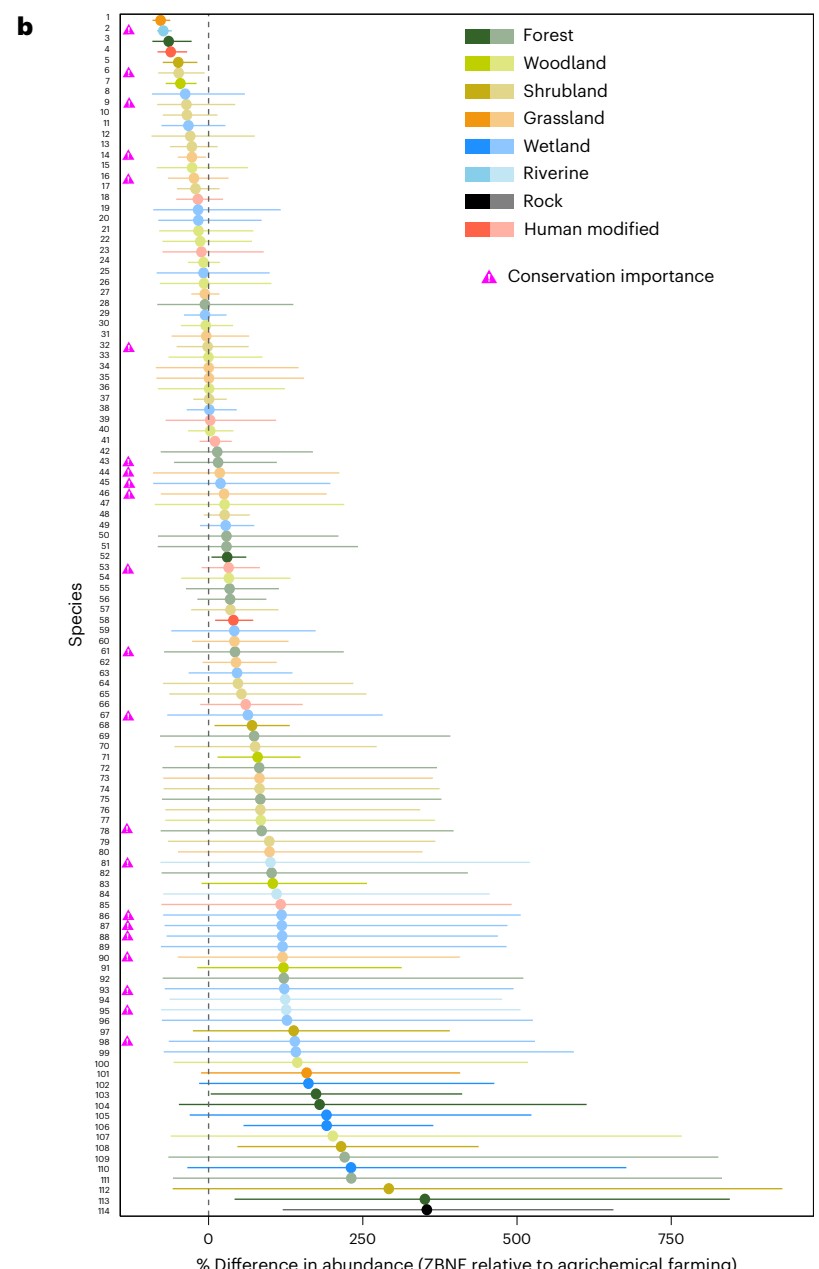

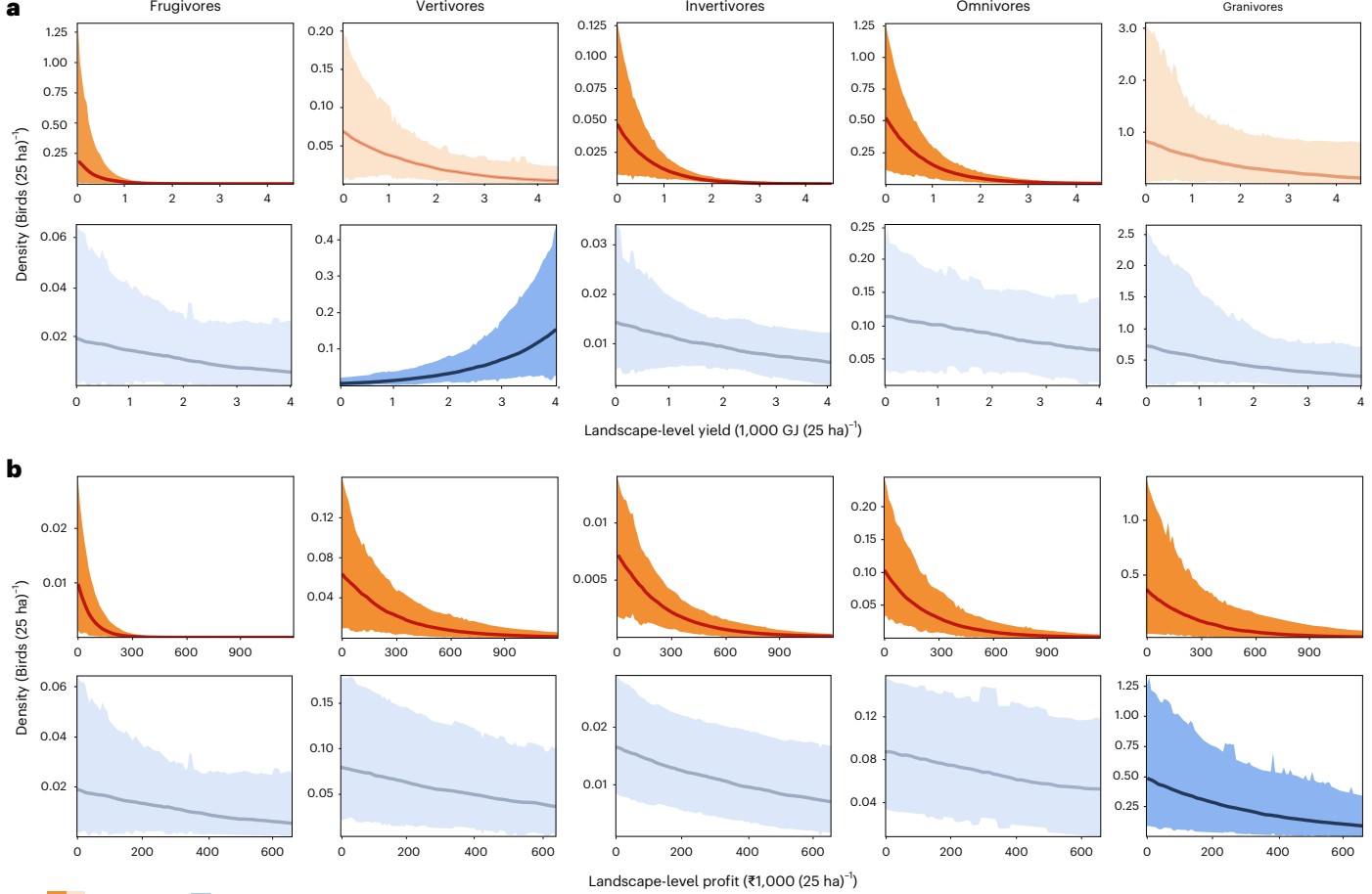

**Fig. 3 | Relationship between landscape-level agricultural productivity and economic profit and bird densities at the trophic-guild level for each farming system. a**, The density of the average frugivore, invertivore and omnivore significantly declined with increasing agricultural productivity in agrichemical systems, whereas in ZBNF systems the densities of the average vertivore significantly increased. For all other guild–farming-system combinations, the density–yield relationships were negative, albeit non-significant.

**b**, In agrichemical systems, the densities of all guilds significantly decreased with increasing profit, but only the average granivore significantly decreased with profit in ZBNF systems. The density–profit relationships were non-significantly negative for all other guilds in ZBNF systems. Shading indicates 95% Bayesian credible intervals, and the amount of opacity denotes the significance level, where full opacity indicates a significant trend (PD > 95%), and 50% opacity indicates a non-significant trend. $n_{species} = 114$, $n_{point counts} = 416$, $n_{landscaspes} = 26$.

boosting the population sizes of these species could have high conservation impact. However, further work is needed to fully understand the relative conservation value of improved management of non-forest habitats and of transitions to ZBNF for these species.

## Trade-offs are less pronounced for ZBNF

We examined the trade-offs between food production or profitability and biodiversity for each farming system by aggregating productivity and profitability estimates at landscape level (25 ha) and correlating these against bird abundances in hierarchical Bayesian models. Again, we fit these models to incorporate variation at both the trophic guild and species levels (see Methods).

In agrichemical systems, the abundance of the average omnivore, invertivore and frugivore significantly declined with increasing landscape-level yield, indicating a direct trade-off between yield and farmland bird biodiversity; the average granivore and vertivore exhibited a negative but statistically non-significant response. Trade-offs in ZBNF systems were less pronounced, and no relationships were significantly negative, indicating that trade-offs are weaker and that yields can be enhanced in this system with lesser loss of bird populations (Fig. 3a, Extended Data Fig. 3a,b and Supplementary Table 9). Similarly, on average, all trophic guilds significantly declined with increasing landscape-level economic profit in agrichemical systems, whereas

only the average granivore declined significantly in ZBNF systems (Fig. 3b, Extended Data Fig. 4a,b and Supplementary Table 9). All other guilds exhibited, on average, negative density–profit relationships in ZBNF systems, but declines were non-significant and less steep than in agrichemical systems (Fig. 3b). At the individual-species level, 47 species had significantly negative density–yield curves in agrichemical systems, and 14 species in ZBNF systems (Extended Data Fig. 3c,d and Supplementary Table 8). Similarly, 34 and 12 species had negative density–profit curves in agrichemical and ZBNF systems, respectively (Extended Data Fig. 4c,d and Supplementary Table 8).

Landscape-level yield and profit are a function of field-level productivity and the amount of embedded (semi-)native vegetation cover, which includes small forest fragments (<0.02 km²), individual native or naturalized trees, and hedges (see Methods). With vegetation cover not differing between ZBNF and agrichemical systems at the 25-ha square level (Wilcoxon–Mann–Whitney test, $P = 0.118$; Supplementary Information 7 and Supplementary Fig. 2), differences in the shapes of the density–productivity curves between the two farming systems are largely driven by differences in field-level management. In agrichemical systems, higher yields are probably achieved through a more intensive use of synthetic pesticides and fertilizers, whereas in ZBNF systems, higher yields may be delivered through greater employment of the ZBNF 'wheels' and carefully attuning them to local conditions.

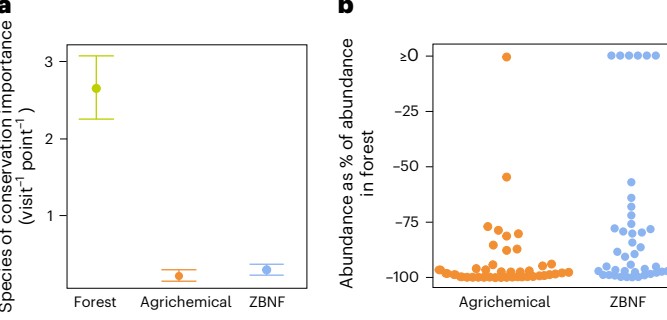

**Fig. 4 | Forests have higher richness and abundance of species of conservation importance than agricultural systems. a**, The average number of species of conservation importance ('Near threatened' or 'Vulnerable' on the IUCN Red List and/or of national conservation priority) is higher in forests than in agricultural landscapes, with no difference between ZBNF and agrichemical systems. Median estimates, with whiskers representing the 95% Bayesian credible intervals. $n_{species}$ = 55, $n_{point counts}$ = 816, $n_{landscaspes}$ = 51. **b**, Estimated median population abundances of the 41 species of conservation importance recorded in forests during this study. Abundances in ZBNF and agrichemical systems are expressed relative to forest populations. Each circle represents a species. $n_{species}$ = 41, $n_{point counts}$ = 816, $n_{landscaspes}$ = 51.

Nonetheless, within each system, increasing landscape-level yield is also a result of a reduction in vegetation cover, which could explain why density–yield trade-offs tended to be more acute for forest-affiliated species (Extended Data Fig. 3c,d).

In ZBNF systems, vertivores significantly increased with increasing landscape-level yield (Fig. 3a and Extended Data Fig. 3b). Similarly, the relationship between density and agricultural productivity was positive for a few species in both farming systems, but this was the case for more species and the strength of the estimated relationships were stronger in ZBNF systems (18 versus 8 species; Extended Data Fig. 3c,d). All of these were species whose primary habitat is not forest, such as waterbirds, and which may have benefited from an increase in open and structurally simpler habitats.

Overall, increases in agricultural and economic productivity negatively affected more species, and the magnitude of the estimated negative effects were greater in agrichemical than in ZBNF systems (Extended Data Figs. 3 and 4, and Supplementary Table 8). Hence, the ZBNF programme was largely able to dampen or occasionally neutralize trade-offs between food production, agrarian livelihoods and bird conservation. Nonetheless, considerably more empirical evidence on the multidimensional performance of ZBNF is needed since the bird density–productivity relationships we observed may not hold in the long-term or reflect broader biodiversity responses.

**Forests hold irreplaceable conservation value**
Ascertaining the scope and limitations of the ZBNF programme to improve regional conservation outcomes also necessitates studying natural ecosystems as a frame of reference. We therefore conducted 100 point counts (each with 4 repeats) in 25 25-ha tropical moist deciduous, dry deciduous and semi-evergreen forest landscapes (Supplementary Fig. 3), and ran three analyses comparing the richness of species of conservation importance, species-level abundances, and community integrity between each farming system and forests (see Methods). Of the 199 bird species we recorded across the study, 85 species were unique to forests (Supplementary Table 6).

On average, more species of conservation importance were found in forests than in ZBNF or agrichemical systems (estimated relative reduction in species richness was −89.0% (−92.0% to −85.7%) and −90.6% (−93.2% to −87.7%), respectively; Fig. 4a). Forest species of conservation importance were considerably less abundant in agricultural landscapes than in forests (Fig. 4b and Supplementary Table 7),

and while ZBNF improved overall community integrity (that is, Bray–Curtis dissimilarity to forest community composition) by 3.32% (0.54 to 5.91%) relative to agrichemical farming, both communities were highly dissimilar to forests (ZBNF 0.94, 95% CI: 0.92 to 0.96; and agrichemical 0.97, 95% CI: 0.96 to 0.99; where 0 indicates the same composition between communities and 1 denotes complete dissimilarity) (see Supplementary Information 8). Thus, while coordinated transitions to ZBNF can revert some of the community declines and shifts associated with forest conversion to agrichemical farming, ZBNF still has very limited value in conserving forest-specialized species.

The wider region has a long history of agricultural encroachment and forest disturbance, including selective logging and replanting[59]. While the effects of selective logging on biodiversity can be relatively subtle[60,61], it tends to lead to a decline or loss of specialist, highly disturbance-sensitive species, leaving behind those species that are more resilient to habitat disturbance and fragmentation[62,63]. This may have been the case in our region. Consequently, with ZBNF's capacity to conserve forest birds already being low in our system, the relative forest bird conservation value of ZBNF (and other agroecological approaches) may be even lower in areas with recent emergence of agricultural activities and that harbour undisturbed natural ecosystems with a high proportion of disturbance-sensitive forest specialists.

Nonetheless, on the basis of our knowledge of the region's avifauna, we consider 34 species we recorded in forests to be disturbance sensitive (Supplementary Table 6). Most of the forests in our study are currently experiencing some form of disturbance (primarily hunting, grazing and harvesting of non-timber products), hence the species' densities would probably be higher in the absence of these disturbances. While the activities are largely permissible with the level of protection the forests are under (namely, Reserved Forests, IUCN Category IV), current practices are deemed unsustainable and modes of governance ineffective[64]. Restoring the integrity of forests and reverting defaunation while supporting the livelihoods of forest-dependent communities will necessitate substantial, multifaceted approaches, including stronger political representation of Indigenous people[65], which has been associated with better forest conservation outcomes in India[66,67].

## Conclusions
Our analyses indicate that the ZBNF programme is a rare example of an agroecological transition delivering win–win outcomes for people and nature without compromising agricultural productivity. This supports the growing interest in adopting ZBNF in numerous other tropical countries[31], although the outcomes of agroecological interventions can be highly context dependent and non-deterministic[24].

Synergies between outcomes have to be actively fostered in most land systems[67]. The ZBNF programme's agricultural extension services and field schools are effective at crafting these, but further benefits could probably be delivered in a more conducive policy environment, for example, via the creation of ZBNF-specific value chains, removal of subsidies in favour of agrichemicals, and integrating ZBNF into landscape-scale conservation planning.

Increases in the profitability of agriculture can incentivize encroachment into natural habitats, with farmers reinvesting profits for additional land clearing[68], and the need to pair agroecological approaches with effective area-based conservation measures has been highlighted[69,70]. Policy interventions may be needed to ensure that ZBNF-induced profitability boosts do not result in further agricultural expansion but instead facilitate the protection and restoration of natural ecosystems. Over the past two decades, India had the second largest net cropland increase globally[71], of which a considerable proportion occurred inside protected areas, with Andhra Pradesh representing a particular hotspot[72]. Simultaneously meeting the area-based conservation targets of the Kunming–Montreal Global Biodiversity Framework and agriculture-related Sustainable Development Goals is particularly

challenging in India[11]. ZBNF can contribute to reconciling them, but it must be complemented with active, equitable conservation of natural habitats and food-system approaches that help to reduce demand, including shifting to healthier diets and reducing crop and food waste.

## Methods

Assessing the socio-economic and biodiversity outcomes of ZBNF necessitates measuring their impact against the counterfactual, that is, what would have happened in the absence of a transition to ZBNF. The counterfactual is inherently unobservable, so robust impact evaluation approaches must be employed to estimate it. In our context, if ZBNF had been randomly assigned to villages, the counterfactual could be estimated by simply comparing the outcomes between ZBNF and agrichemical (non-ZBNF) villages, and we would assume that the non-ZBNF villages represented how the ZBNF villages would look if they had not transitioned. However, the non-randomized uptake of ZBNF means that care must be taken to control for confounding factors that affect both ZBNF programme participation and outcomes (that is, rival explanations; for instance, a farm with higher biodiversity might be more likely to enrol in ZBNF). We aimed to minimize the differences in biophysical and socio-economic covariates thought to be confounders between ZBNF and agrichemical sites by careful site selection in the field and statistical matching. Matching entails a range of statistical methods aiming to improve causal inference of subsequent analyses by identifying, and sometimes assigning variable statistical weights to, sampling units where balance of predefined measurable characteristics that influence both the likelihood of a unit being subjected to an intervention and the outcome of interest[73,74] is achieved between intervention and control units (see Supplementary Information 5). We quantified impacts on yield and profit using g-computation and fitted five different Bayesian models to examine bird biodiversity outcomes (see below and Supplementary Fig. 4).

We performed all statistical analyses in R v.4.3.1. For each analysis, we tested whether assumptions for normality and homogeneity of variances were met, checked for overdispersion using the DHARMa[75] package and checked for spatial autocorrelation by computing Moran's *I* coefficient using the ape[76] package.

### Sample landscape ('square') selection

We used ZBNF programme documentation from governmental and non-governmental sources to identify villages where farmers are almost exclusively practicing ZBNF (>80% of farmers) across north-east Andhra Pradesh, India. Of 206 villages, we removed those where ZBNF had been practised for less than 4 years, where it was not possible to find a 500 × 500 m area that is exclusively under ZBNF, and villages where this square would have been closer than 3 km from land use of a different category (see Supplementary Information 2 for a full justification for selecting this square size). We also required 75% of the farmed area in the 5-km radius surrounding the centre of each square to be managed under ZBNF, and we removed villages with extreme (that is, unrepresentative) elevation, slope, soil type and/or climatic attributes. We examined the amount of natural and semi-natural vegetation embedded in the square and surrounding landscape in Google Earth Engine. We then visited the remaining 28 villages and, after ground-truthing the area cover of ZBNF, were left with 13 ZBNF 500 m × 500 m squares. We then selected 13 agrichemical and 25 forest squares in the region with similar attributes of the above-described biophysical variables (see Supplementary Fig. 5), and with agrichemical squares spanning the same spectrum of embedded (semi-)native vegetation cover as the ZBNF squares (which encompassed small forest fragments (<0.02 km²), individual native or naturalized trees, and hedges; see Supplementary Information 2 and Supplementary Fig. 2 for details). All study squares were at least 5 km from one another. On average, the farmland squares contained 174 (±82.6 s.d.) fields (191.2 ± 93.01 and 156.8 ± 70.1 for ZBNF and agrichemical squares, respectively). We surveyed forest habitats

as a reference community, because forests were the dominant ecosystem replaced by agriculture and are still the main remaining natural ecosystem in the region[59,77,78].

### Point and field selection

Within each square we positioned four points, evenly spaced 200 m apart and 150 m from the square edges. For the agricultural sites, we interviewed the farmers managing the agricultural fields upon which the points fell, as well as up to two more farmers per square to obtain information on yield, revenue and costs. We also conducted our avian counts from these points.

### Outcome data

**Yield and economic profit.** We conducted 128 structured interviews of farmers between December 2022 and February 2023 to elicit information on the agronomic factors needed to estimate crop yields and economic profit. Before participating in the study, all farmers gave informed consent. The interviews asked for detailed information about each identified field over a 1-year recall period, thus capturing the output and costs associated with all crops successively grown on the same field per annum (see Supplementary Information 3). This included information on cropping cycles, method of land preparation and management, input and labour costs, and equipment purchases and rentals. We constructed simple crop lifecycle models and, to allow for comparability across crop types, estimated the energetic value of each harvest and subsequently of each field per annum (GJ ha⁻¹). Similarly, we used wholesale state-level prices and farmers' self-reported input costs and yield to estimate the profit per field (₹ ha⁻¹ year⁻¹). We quantified the amount of native and semi-native vegetation (% area) in each 25-ha square using a handheld GPS and satellite imagery, and estimated square-level productivity and profit by multiplying the respective mean estimates by the area under cropland in a square. See Supplementary Information 3 for details.

Across the 26 agricultural squares, we obtained information on 128 fields (66 ZBNF fields and 62 agrichemical fields, where 5.23 ± 0.83 ZBNF and 4.62 ± 1.26 agrichemical farmers were interviewed per square) and obtained yield information on 206 harvests (115 ZBNF harvests and 91 agrichemical harvests). Most farmers grew multiple crops per year on the same piece of land.

**Bird densities.** We sampled a total of 51 squares, each containing 4 point locations: 25 squares in the continuously forested areas, 13 in ZBNF and 13 in agrichemical farming. Following ref. 79, we conducted 10-min point counts with no settling-in period at each point location in both the winter (December–March) and summer (April–June) seasons over 2 years (2021/2022 and 2022/2023). Hence, we visited each point on 4 separate occasions, totalling 816 repeats (51 squares × 4 points × 4 repeats) and 8,160 min of observation (160 min per square) across all sites.

We completed all counts between 15 min before and 3 h after sunrise in fine weather (no strong winds or heavy rain). At each point, we estimated the distance to individuals we recorded from visual and/or aural cues using a laser rangefinder. We recorded birds that flushed while approaching the point (noting the distance to their initial location), but birds we only saw in flight and birds that flew in during the count period were excluded (following ref. 80). We identified individuals and assigned them to species according to BirdLife International taxonomy. We removed observations greater than 100 m from the point and excluded largely aerial species (see Supplementary Information 4). To account for varying detectability of different species in different habitat types, we used distance-sampling methods. These models enabled us to estimate bird densities (in contrast to other techniques such as occupancy modelling[80]), adjusted for how detectable each species is in different environments. For the 52 species that had at least 40 observations, we fitted species-specific detection functions (where

the probability of detecting an individual is a function of the distance from the observer). We also grouped all species on the basis of detectability (underpinned by functional trait data) and ran group-specific detection functions. We took functional trait data from AVONET[81] and formed 18 detectability groups on the basis of taxonomic, dietary and primary lifestyle characteristics. We used these group-specific detection functions for the 147 species with fewer than 40 observations each.

For each species, or detectability group, we fitted detection functions using the mrds[82] package. For robustness, we considered three models for each: (1) a single detection function with the proportion of closed habitat around each point ($r = 100$ m) as a covariate (as this is likely to affect detectability); (2) a single detection function with no covariates; and (3) separate detection functions for forest and/or farmland sites, provided there were at least 40 observations in each. For the group-level detection functions, we included species as a covariate in each model.

In each case, we tested half-normal and hazard-rate key functions with cosine, hermite polynomial and simple polynomial adjustment terms. We obtained a list of feasible detection functions (that is, offering a good fit). We preferred (3) over (1) and (2) if it offered a good fit for a given species or detectability group at each site; otherwise, we used the small-sample corrected Akaike Information Criterion (AICc) for model selection (see Supplementary Information 4 for details). Using these models, we obtained 'effective area surveyed' estimates for each species at each point count location (either from the relevant species-specific model or the detectability-group model), which act as a proxy for detectability. These values were then used as offsets in our main models (that is, the multispecies count models).

### ZBNF's impact on yield and profit

**Matching.** We conducted matching at the harvest level when comparing yield (thus, a particular crop grown at a particular time, remembering that multiple different crops could be grown on one field within a year) and at the field level when comparing economic profit (since it was not possible to estimate profit for each harvest but only as a per annum estimate for a given field).

Before matching, we examined the data for outliers, which led to the removal of four tapioca harvests when examining ZBNF's impact on yield (as these had implausibly high yield values and tapioca is also extremely rarely grown in the region) and left us with information on 115 ZBNF harvests and 87 agrichemical harvests. As a robustness check, we repeated the analysis with the implausible tapioca harvests included, and results were not substantially different (see Supplementary Information 6). For the profit analysis preprocessing, we removed seven fields that had estimated profits of over ₹25,000 per hectare, and we were thus left with 66 ZBNF fields and 55 agrichemical fields (see Supplementary Information 6 for a full justification).

We checked for collinearity of covariates by calculating the generalized variance inflation factor (GVIF; using the VIF function in the car package in R[83]) of all covariates, but since no variables had a GVIF >4 and a Pearson's correlation coefficient >0.7, no covariates had to be removed[84]. In general, the covariates included in a matching analysis should be underpinned by a theory of change[73,74]. We collected data on observable characteristics that we believed to capture the key confounding factors in an evaluation of the ZBNF programme's impacts on food production, profit and bird communities (where, for time-varying confounders, we used data from the year before the ZBNF transition, that is, 2015). When assessing ZBNF's impact on yield, we matched on agricultural suitability, crop type, travel time to cities with a population >50,000, (semi-)native vegetation cover, the proportion of harvest sold, land ownership and whether the harvest was irrigated or rainfed (see Supplementary Table 2 for a full description of each matching variable and justifications for their inclusion). When ascertaining ZBNF's impact on economic profit, we matched on agricultural suitability (see Supplementary Information 5 for justifications and

Supplementary Tables 3 and 5 for robustness checks). Given our efforts to find agrichemical squares in reasonable proximity and with similar characteristics to our ZBNF squares, our samples were relatively similar even before matching (Supplementary Figs. 6 and 7).

We conducted multiple matching runs with variations in the algorithm, the distance metric (which quantifies the similarity between units) and whether matching was conducted with or without replacement (see Supplementary Information 5 and Supplementary Table 3 for details). Doing so is generally advised[85] and allowed us to find the method yielding the best matches, and also ensured that our results are robust to the matching specifications employed. Using the MatchIt[86] R package, we assessed the covariate balance between the two farming practices using the 'standardized difference in means' (SDiM) for all runs, where we considered a covariate adequately balanced if it had an SDiM below 0.25 (refs. 73,74,85,87,88). Matching runs where more than two covariates remained imbalanced and/or where <30% of the agrichemical samples were matched (implying that they are less likely to be representative of the target population) were not used for further analysis.

At the harvest level (that is, matching for analysing ZBNF's impact on yield), three matching runs out of ten met these criteria, namely, nearest-neighbour Mahalanobis matching with replacement, genetic matching with replacement matched on the covariates, and genetic matching with replacement matched on the covariates and each harvest's propensity score. The latter resulted in the best balance with the highest sample size (Supplementary Fig. 6) and was thus considered as our main analysis, although we conducted post-matching analyses for the other two matching runs as well (see Supplementary Information 5 and 6). At the field level (that is, before the profit analysis), full matching with Mahalanobis distance achieved complete balance with the highest sample size (Supplementary Fig. 7) and thus represented the main analysis (although again, we also conducted analyses for other matching runs; see Supplementary Information 5 and 6). Mahalanobis distance matching calculates how many standard deviations a unit is from the mean of other units, whereas genetic matching is based on a generalization of this and entails iterative searches to maximize the balance of covariates between treatment and control.

**Analytical framework.** We first square-root transformed the harvest-level yield data (to achieve normality) and then ran linear mixed-effects models (using the lme4[89] package) to assess the effect of the ZBNF programme on yield. Besides farming practice and the matching weights, we included the percentage of habitat patches as a fixed effect (as its SDiM was slightly above 0.25) and square identity as a random term. When economic profit was the response variable, we fitted farming practice as a fixed effect and square identity as a random term. In all instances, we used g-computation[90] to estimate the effect of ZBNF and we estimated bias-corrected accelerated (BCa) bootstrap confidence intervals using the boot[91] package with 9,999 replications. G-computation involved computing each sampling unit's predicted values of the outcome, setting their treatment status to 'treated' (that is, ZBNF), and then again for the control (that is, agrichemical), which left us with two predicted outcome values for each unit. We computed the mean of each of the estimated potential outcomes across the entire sample. This left us with two average estimated potential outcomes for each farming system and the contrast of these is the estimate of the effect of ZBNF[90].

We subjected our designs to 14 robustness checks, such as repeating our analysis for rice harvests from the main growing season only and conducting separate analyses for different subregions with distinct biophysical and socio-economic characteristics (described in Supplementary Information 6).

### ZBNF's impact on birds

For all below-described Bayesian models, we specified zero-centred diffuse priors for the intercept and beta parameters (normally distributed

with a mean of 0 and standard deviation of 10), and ran the models with 4 chains, each with 1,000 warm-up iterations and 2,000 post-warm-up sampling iterations, totalling 8,000 posterior iterations. Model convergence was verified by examining trace plots and the 'Rhat' statistic and model fit, and adequacy was evaluated using posterior-predictive plots. We performed all modelling using the brms package[92] as an interface to the Bayesian inference engine 'Stan'.

**Matching.** We matched at the point level on elevation, temperature, and the proportion of native and semi-native vegetation patches (rainfall was excluded due to collinearity with elevation). We conducted the same matching runs as described above, where full matching with Mahalanobis distance performed best (complete balance with highest sample size; Supplementary Fig. 8).

**Multispecies count model.** We applied a hierarchical zero-inflated Poisson count model to estimate individual-species counts per point per visit, $Count_{i,p,v}$, where $i$ indexes species, $p$ indexes points and $v$ indexes visits. We modelled zero inflation ($\varphi$) using a Bernoulli distribution, with farming practice (FP: ZBNF or agrichemical) as a fixed effect and species as a hierarchical varying intercept (indexed by $i$). For the Poisson-distributed count model, we modelled the number of individuals of a species recorded during a given visit to a point weighted by the matching weights (applied as frequency weights) as a function of the fixed effect of farming practice (FP) and trophic niche (TN: granivore, frugivore, omnivore, invertivore and vertivore; obtained from AVONET[81]), as well as their interaction. We further allowed the intercept and slope of farming practice effect to vary by species ($v_{species[i]}$, indexed by $i$) and included a hierarchical structure of points nested within squares to account for our sampling design ($u_{point/square[p]}$, indexed by $p$). The effective area (EA) surveyed was included as an offset.

$$Count_{i,p,v} \sim \begin{cases} Poisson\left(\lambda_{i,p}\right) & \text{with prob } 1-\varphi \\ 0 & \text{with prob } \varphi \end{cases}$$

$$logit\left(\varphi_i\right) = v_{species[i]} + \beta_{0,i} + \beta_1 FP_i + \in \qquad (1)$$

$$log\left(\lambda_{i,p}\right) = \beta_{0,i,p} + u_{point/square[p]} + v_{species[i]} + \beta_1 FP_i +$$
$$\beta_2 TN + \beta_3 FP \cdot TN + offset\left(log(EA)\right) + \in$$

To ascertain the effect of the ZBNF transition across all farmland birds at the guild and species levels, we extracted the posterior draws from a newly created square × species matrix, where we held species and guild-level effects constant as appropriate.

We paired abundance estimates of species with data on their characteristics. We regarded a species to be of conservation importance if it was assessed as 'Near threatened' or 'Vulnerable' under the IUCN Red List and/or if it was considered of 'High' or 'Moderate' conservation priority under the State of India's Birds 2023 report[52]. Information on the main habitat type a given species is predominantly associated with was extracted from AVONET[81]. We also conducted a supplementary analysis where we additionally included species' primary lifestyle (terrestrial, insessorial or generalist) and body mass (in log-transformed grams), again obtained from AVONET[81], as fixed effects, each interacted with farming practice, in the above-described model.

### Trade-offs between bird densities and landscape-level productivity and profit

**Multispecies count model.** We fitted a zero-inflated hierarchical Bayesian model with a Poisson distribution, here including observations from the forest, ZBNF and agrichemical sites, and included square-level yield or, in a separate model, square-level economic profit (see Supplementary Information 3), as a fixed effect to examine

guild- and species-level density–productivity relationships and to compare densities between forest and each farming system (see below). Having three land-use types meant that we could not conduct matching. Instead, we included elevation (EL) and temperature (TP) as covariates (see Supplementary Information 7 for a discussion on vegetation cover). Counts (per species, per point, per visit) were modelled using a zero-inflated Poisson distribution. Since agricultural productivity is only applicable to farmland sites and we wanted to ascertain farming-practice-specific effects of yield (and profit), we coded our site types as binary terms (ZBNF, Chemical, with a zero in both terms denoting forest); likewise, we included the yield (or profit) of each practice (Chemicalyield and ZBNFyield), all of which were allowed to vary by trophic niche (TN). We included a random effect structure of points nested within squares (point/square [$p$] indexed by $p$), and we allowed the intercept and land-use type ($\beta_1 ZBNF_i$, $\beta_2 Chemical_i$) and yield ($\beta_6 ZBNFyield_i$, $\beta_7 Chemicalyield_i$, or profit) to vary by species ($v_{species[i]}$ indexed by $i$). To construct density–productivity curves, we then extrapolated the point count estimates to obtain density estimates at the square level.

$$Count_{i,j,k} \sim \begin{cases} Poisson\left(\lambda_{i,j,k}\right) & \text{with prob } \varphi \\ 0 & \text{with prop } 1-\varphi \end{cases}$$

$$logit\left(\varphi_i\right) = \alpha_{,i} + v_{species[i]} + \beta_1 Chemical + \beta_2 TNk + \in \qquad (2)$$

$$log\left(\mu_{ijk}\right) = u_{point/square[p]} + v_{species[i]} + \beta_0 + \beta_1 ZBNF_i$$
$$+\beta_2 Chemical_i + \beta_3 TN + \beta_4 ZBNF \cdot TN + \beta_5 Chemical \cdot TN + \beta_6 ZBNFyield_i$$
$$+\beta_7 Chemicalyield_i + \beta_8 ZBNFyield \cdot TN + \beta_9 Chemicalyield \cdot TN + EL$$
$$+TP + offset\left(log(EA)\right) + \in$$

### Comparison to forests

**Species abundances.** We extracted species-level abundance estimates from the model described in equation (2); for the subset of species of conservation importance recorded in forests, we expressed abundance estimates in ZBNF and agrichemical systems relative to those in forests.

**Richness of species of conservation importance.** We compared the number of species of conservation importance recorded during a given visit to a point between forests, ZBNF and agrichemical systems. We fitted a Bayesian model with a zero-inflated Poisson distribution, with the zero inflation being a function of the land system type (LT: ZBNF, agrichemical and forest) and the Poisson counts being a function of land system type and a random effect structure of points nested within squares ($u_{point/square[p]}$, indexed by $p$). Here, the effective area (EA) surveyed was averaged across all species at a given point.

$$Richness_{i,p,v} \sim \begin{cases} Poisson\left(\lambda_{i,p}\right) & \text{with prob } \varphi \\ 0 & \text{with prob } 1-\varphi \end{cases}$$

$$logit\left(\varphi_i\right) = \beta_{0,i} + \beta_1 LT + \in \qquad (3)$$

$$log\left(\lambda_{i,p}\right) = u_{point/square[p]} + \beta_0 + \beta_1 LT + offset\left(log(EA)\right) + \in$$

**Community integrity.** We calculated bird community integrity using the abundance-based Bray–Curtis similarity index (calculated using the vegan[93] package), where we quantified the difference in species composition between each forest point and each ZBNF or agrichemical point (having summed observations across visits to each point). We then modelled the Bray–Curtis similarity index between farming practices using a Bayesian hierarchical model with a zero–one inflated beta distribution to account for the left-skewness and the bounded nature

of the data (in the range of 0–1). The zero–one inflated model specifically models the observed Bray–Curtis similarity index ($y_{af}$, where $a$ and $f$ index agriculture–ZBNF and agrichemical–and forest points) as the non-zero-or-one index ($\mu$), the shape parameter of the non-zero-or-one beta distributed index ($\phi$), the zero-and-one probability ($\alpha$) and the conditional one probability ($\gamma$). In this model, we included farming practice (FP) as a fixed effect, a nested random effect structure indexing each agricultural point nested within a square ($u_{\text{point/square}[a]}$), and further included the forest point ($v_{\text{forest}[f]}$) a given agricultural point was compared to as an un-nested random intercept for the non-one-or-zero index values, the one and zero probability and the conditional one probability. We included the same matching weights as frequency weights as above. As a robustness check, we also repeated the analysis at the square level (see Supplementary Information 8).

$$f(y_{af}) \sim \begin{cases} \alpha_{af}(1-\gamma_{af}) & \text{if } y = 0 \\ \alpha_{af} * \gamma_{af} & \text{if } y = 1 \\ (1-\alpha_{af}) * \text{Beta}(\mu,\phi) & \text{if } y \notin \{0,1\} \end{cases}$$

$$\text{logit}(\alpha_{af}) = u_{\text{point/square}[a]} + v_{\text{forest}[f]} + \beta_0 + \beta_1\text{FP} + \in_{af}$$

$$\text{logit}(\gamma_{af}) = u_{\text{point/square}[a]} + v_{\text{forest}[f]} + \beta_0 + \beta_1\text{FP} + \in_{af} \qquad (4)$$

$$\text{logit}(\mu) = u_{\text{point/square}[a]} + v_{\text{forest}[f]} + \beta_0 + \beta_1\text{FP} + \in_{af}$$

$$\log(\phi) \sim \beta_0$$

### Reporting summary

Further information on research design is available in the Nature Portfolio Reporting Summary linked to this article.

## Data availability

The interview and bird data used in this study are available via Zenodo at https://doi.org/10.5281/zenodo.16687021 (ref. 94).

## Code availability

The R code used in this study is available via GitHub at https://github.com/irisberger/ZBNF (ref. 95).

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

## Acknowledgements

I.B. was supported by the Whitten Studentship, School of Biological Sciences Fieldwork Fund, Tim Whitmore Zoology Fund, Hitchcock Fund and Corpus Christi College Research Fund at the University of Cambridge. The funders had no role in study design, data collection and analysis, decision to publish or preparation of the manuscript. We are immensely grateful to all the farmers who participated in this study, to R. Bandi and F. Tampal from WWF-India, and to Rythu Sadhikara Samstha staff for their support in the field. This paper is dedicated to the late Parthib Basu, without whose energy, passion and scientific excellence it would never have been possible.

## Author contributions

I.B. and L.V.D. conceived the project idea and designed the data collection; I.B., A.K., V.R., V.J. and S.R.N. collected and processed the data, with logistical support and field training from P.B.; I.B. and O.M. conducted the analyses with contributions from H.S.W., D.P.E. and L.V.D.; I.B. drafted the manuscript and all authors contributed to the revision of the manuscript.

## Competing interests

The authors declare no competing interests, but we note for transparency that Rythu Sadhikara Samstha (the agency in charge of rolling out the ZBNF programme) provided us with information on where ZBNF is widely practiced. However, they had no influence on the final site selection and data collection process.

## Additional information

**Extended data** is available for this paper at https://doi.org/10.1038/s41559-025-02849-7.

**Correspondence and requests for materials** should be addressed to Iris Berger.

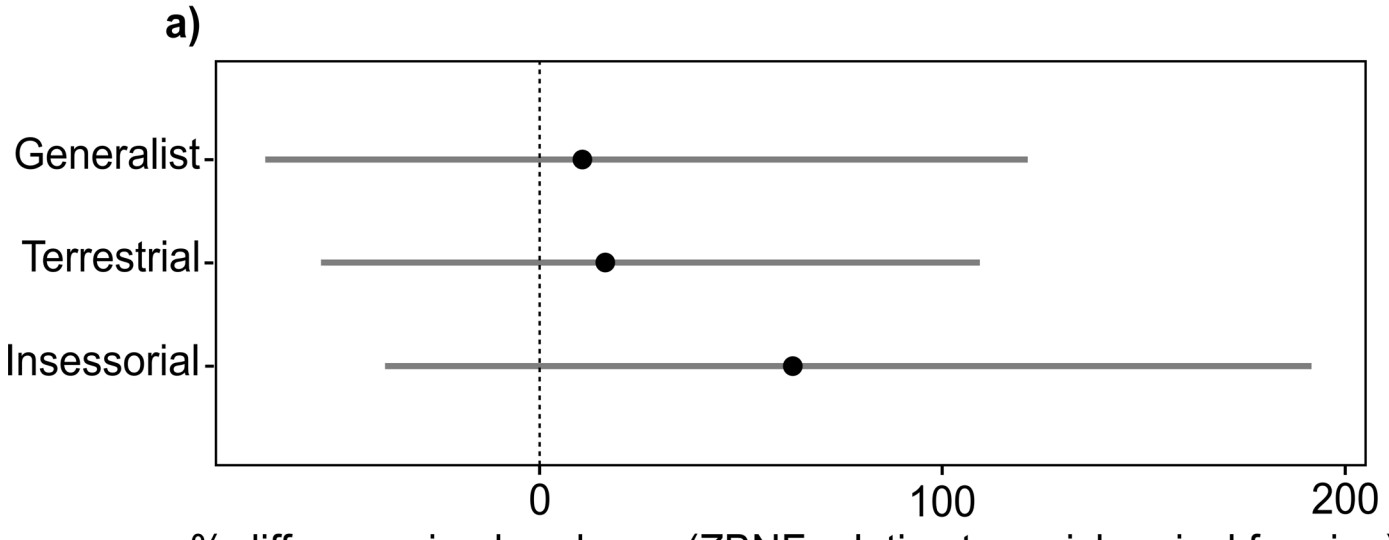

**Extended Data Fig. 1 | Responses to the ZBNF programme are neither driven by species' primary lifestyle nor body mass.** The responses are expressed as a percentage change in abundance compared to agrichemical farming, where the median estimates and their 95% Bayesian credible intervals are shown. **a**) On average, changes in abundance did not significantly differ between primary lifestyle classes. Species with a primary terrestrial lifestyle spend the majority of their time on the ground, those with an insessorial lifestyle spend most of their time perched above the ground (for example on trees or posts), and generalists spend their time in different lifestyle classes. Four species were removed prior to the analysis (namely *Glareola maldivarum*, *Sterna aurantia*, *Anhinga melanogaster*, and *Microcarbo niger*) as they had a primary lifestyle that was aerial or aquatic, and the number of observations in each these classes were too few to model. **b**) The estimated relative change in abundance did not significantly vary along the body mass continuum. $n_{species} = 114$, $n_{point counts} = 256$, $n_{landscaspes} = 26$.

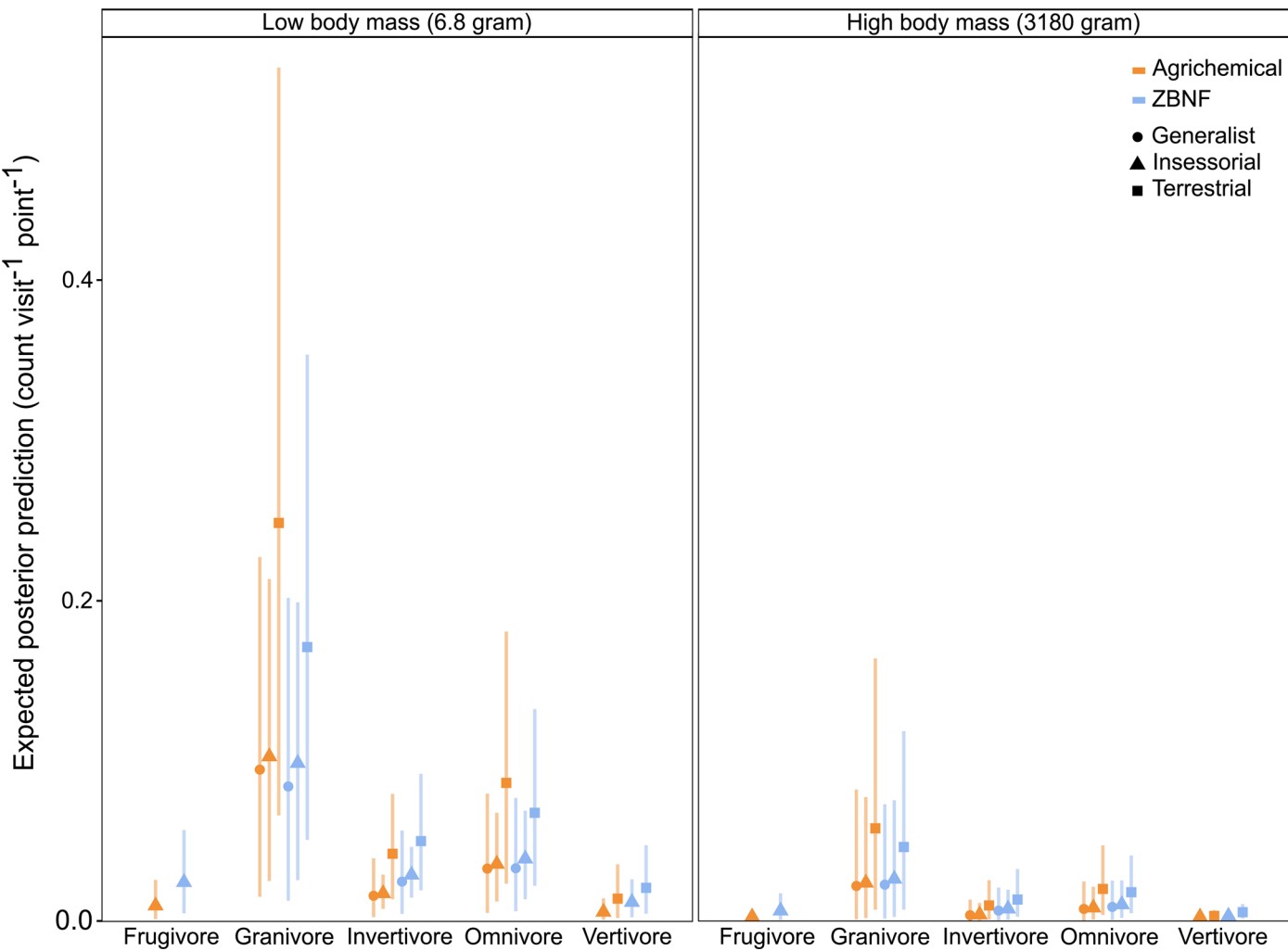

**Extended Data Fig. 2 | Variation across trophic guild, primary lifestyle, and body mass on abundance in each farming system.** Expected median posterior predictions and 95% Bayesian credible intervals for each group are depicted. Body mass values represent the lowest and highest recorded body mass values. $n_{species} = 114$, $n_{point\,counts} = 416$, $n_{landscaspes} = 26$.

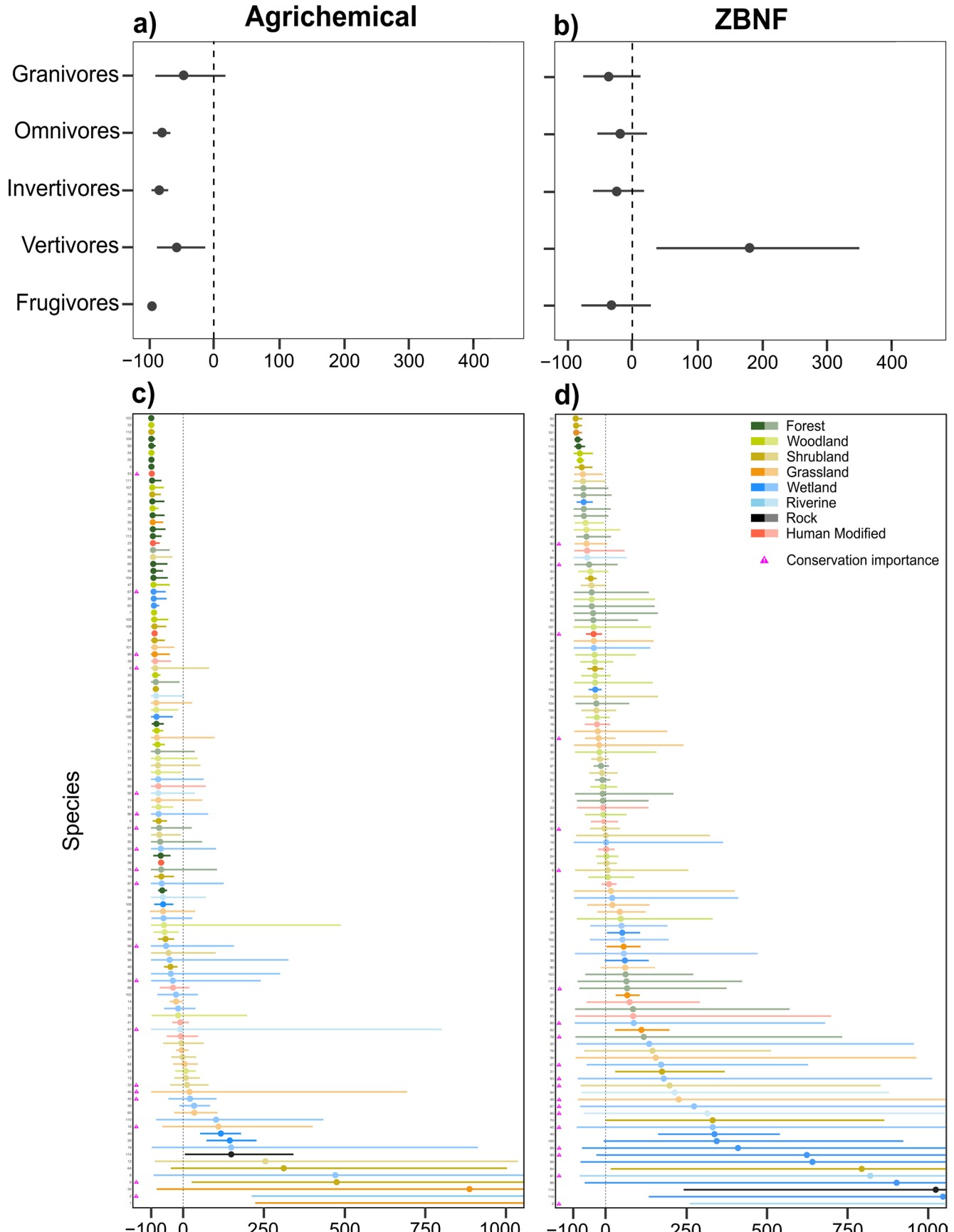

**Extended Data Fig. 3 | See next page for caption.**

**Extended Data Fig. 3 | Farming system-wise relationships between increasing landscape-level agricultural productivity and bird abundances. a)** The abundance of the average omnivore, invertivore, and frugivore declined with increasing yield in agrichemical systems, whereas **b)** the abundance of the average vertivore increased with increasing yield in ZBNF systems. **c)** In agrichemical systems, 47 species declined, 8 increased, and 59 had non-significant responses to increasing landscape-level yield, whereas **d)** 14 species declined, 18 species increased, and 82 species exhibited non-significant responses to increasing yield in ZBNF systems. The responses are expressed as a percentage change in abundance with a one unit increase in yield, where the median estimates and their 95% Bayesian credible intervals are shown. The colour of the error bars depicts a species' the primary habitat (obtained from AVONET): forest (tall tree-dominated vegetation with more or less closed canopy), woodland (medium stature tree-dominated habitats), shrubland (low stature bushy habitats), grassland (open dry to moist grass-dominated landscapes), wetland (wide range of freshwater aquatic habitats including lakes, marshes, swamps and reedbeds), riverine (associated with rivers and streams), rock (rocky substrate typically with no or very little vegetation), and human modified (urban landscapes, intensive agriculture, gardens). Error bars' opacity reflects the proportion of the posterior distribution (PD) that shares the same direction of response as the median (that is, confidence in the response), where 100% opacity reflects PD > 95%, and 50% opacity reflects no significant change. Species of conservation importance (IUCN red-list and/or of national conservation priority) are highlighted. $n_{species} = 114$, $n_{point counts} = 416$, $n_{landscaspes} = 26$.

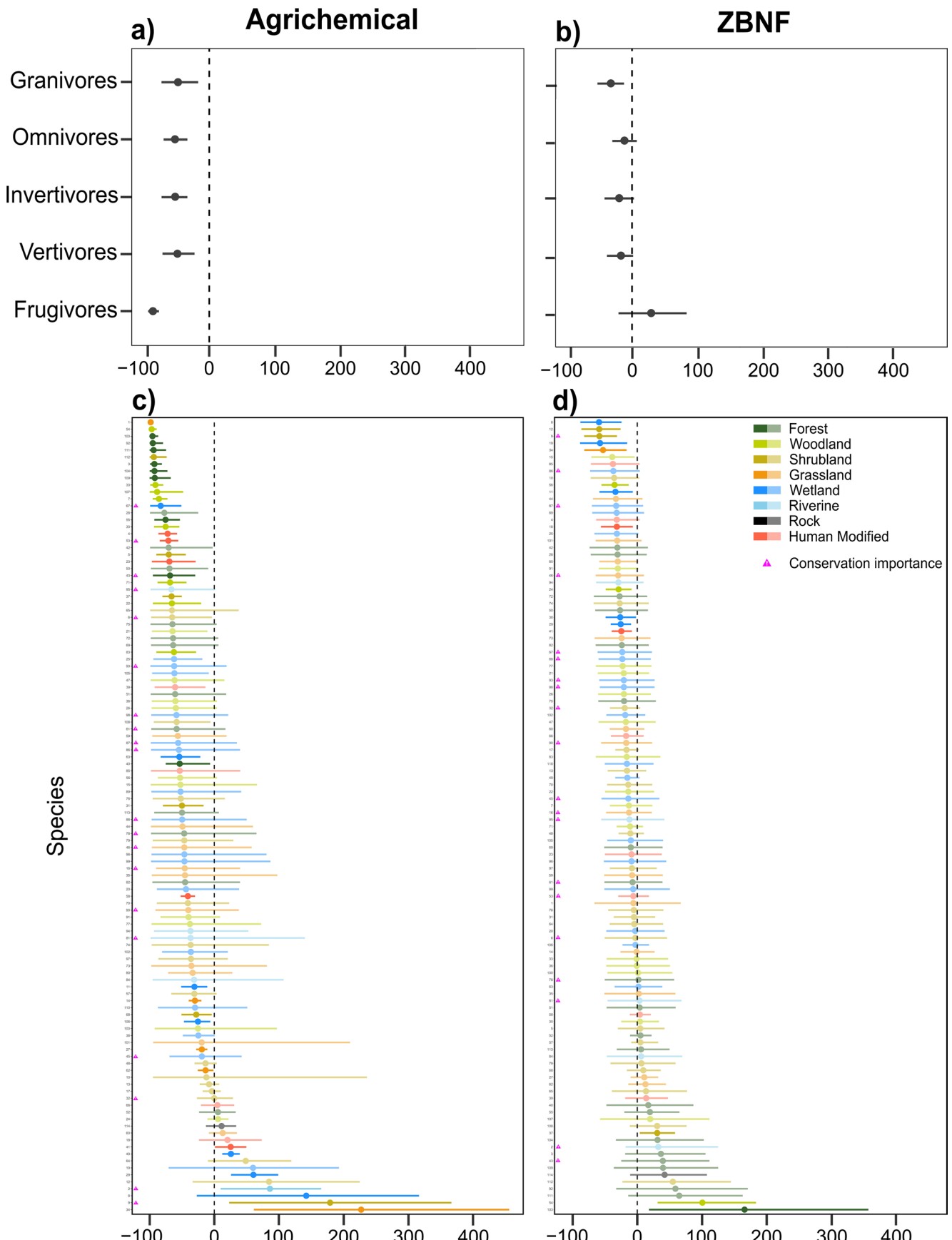

**Extended Data Fig. 4 | See next page for caption.**

**Extended Data Fig. 4 | Farming system-wise relationships between increasing landscape-level economic profit and bird abundances. a)** The abundance of the average species in all trophic guilds declined with increasing yield in agrichemical systems, whereas **b)** only the abundance of the average granivore declined with increasing profit in ZBNF systems. **c)** In agrichemical systems, 34 species declined, 7 increased, and 73 had non-significant responses to increasing landscape-level profit, whereas **d)** 12 species declined, 3 species increased, and 99 species exhibited non-significant responses to increasing profit in ZBNF systems. The responses are expressed as a percentage change in abundance with a one unit increase in profit, where the median estimates and their 95% Bayesian credible intervals are shown. The colour of the error bars depicts a species' the primary habitat (obtained from AVONET): forest

(tall tree-dominated vegetation with more or less closed canopy), woodland (medium stature tree-dominated habitats), shrubland (low stature bushy habitats), grassland (open dry to moist grass-dominated landscapes), wetland (wide range of freshwater aquatic habitats including lakes, marshes, swamps and reedbeds), riverine (associated with rivers and streams), rock (rocky substrate typically with no or very little vegetation), and human modified (urban landscapes, intensive agriculture, gardens). Error bars' opacity reflects the proportion of the posterior distribution (PD) that shares the same direction of response as the median (that is, confidence in the response), where 100% opacity reflects PD > 95%, and 50% opacity reflects no significant change. Species of conservation importance (IUCN red-list and/or of national conservation priority) are highlighted. $n_{species} = 114$, $n_{point\ counts} = 416$, $n_{landscaspes} = 26$.

# Reporting Summary

## Statistics

For all statistical analyses, confirm that the following items are present in the figure legend, table legend, main text, or Methods section.

| n/a | Confirmed | |
|---|---|---|
| ☐ | ☒ | The exact sample size (*n*) for each experimental group/condition, given as a discrete number and unit of measurement |
| ☐ | ☒ | A statement on whether measurements were taken from distinct samples or whether the same sample was measured repeatedly |
| ☐ | ☒ | The statistical test(s) used AND whether they are one- or two-sided *Only common tests should be described solely by name; describe more complex techniques in the Methods section.* |
| ☐ | ☒ | A description of all covariates tested |
| ☐ | ☒ | A description of any assumptions or corrections, such as tests of normality and adjustment for multiple comparisons |
| ☐ | ☒ | A full description of the statistical parameters including central tendency (e.g. means) or other basic estimates (e.g. regression coefficient) AND variation (e.g. standard deviation) or associated estimates of uncertainty (e.g. confidence intervals) |
| ☐ | ☒ | For null hypothesis testing, the test statistic (e.g. *F*, *t*, *r*) with confidence intervals, effect sizes, degrees of freedom and *P* value noted *Give P values as exact values whenever suitable.* |
| ☐ | ☒ | For Bayesian analysis, information on the choice of priors and Markov chain Monte Carlo settings |
| ☐ | ☒ | For hierarchical and complex designs, identification of the appropriate level for tests and full reporting of outcomes |
| ☐ | ☒ | Estimates of effect sizes (e.g. Cohen's *d*, Pearson's *r*), indicating how they were calculated |

*Our web collection on statistics for biologists contains articles on many of the points above.*

## Software and code

Policy information about availability of computer code

| | |
|---|---|
| Data collection | QGIS version 3.38.3 and Google Earth Engine version 7.3 were used to aid in the selection of field sites. The outcome, primary predictor, and some covariate data were obtained through field studies. Area measurements of different land-uses were conducted in QGIS version 3.38.3 (see Supplementary Information 3). The other covariate data were downloaded from various sources online (see Supplementary Information 2 and Supplementary Table 2) and processed in Google Earth Engine version 7.3.All data were collated and cleaned in R version 4.3.1. |
| Data analysis | All analyses were carried out in R version 4.3.1. The key packages used were: "ape", "boot", "brms", "car", "DHARMa", "dplyr", "ggplot2", "lme4", "MatchIt", "mrds", "tidybayes", and "vegan".Aesthetic figure edits were conducted using Inkscape version 1.4. The code is available on GitHub: https://github.com/irisberger/ZBNF |

For manuscripts utilizing custom algorithms or software that are central to the research but not yet described in published literature, software must be made available to editors and reviewers. We strongly encourage code deposition in a community repository (e.g. GitHub). See the Nature Portfolio guidelines for submitting code & software for further information.

## Data

Policy information about availability of data

All manuscripts must include a data availability statement. This statement should provide the following information, where applicable:

- Accession codes, unique identifiers, or web links for publicly available datasets
- A description of any restrictions on data availability
- For clinical datasets or third party data, please ensure that the statement adheres to our policy

> The interview and bird data used in this study are available via Zenodo at: 10.5281/zenodo.16687021

## Research involving human participants, their data, or biological material

Policy information about studies with human participants or human data. See also policy information about sex, gender (identity/presentation), and sexual orientation and race, ethnicity and racism.

| | |
|---|---|
| Reporting on sex and gender | Data on sex and gender were not collected. |
| Reporting on race, ethnicity, or other socially relevant groupings | For our main analyses, we sampled from 'plain' and 'tribal' areas equally. Traditional and indigenous farming practices, low accessibility, and a high proportion of subsistence farmers characterise "tribal" areas. In contrast, 'plain' areas, which are nearer to the coast and better connected to agricultural markets, are dominated by farming systems that have adopted high use of pesticides, fertilisers, irrigation, and agricultural credit. However, as a robustness check, we also conducted separate analyses for 'tribal' and "plain" areas (see Supplementary Information 6). |
| Population characteristics | The participants were smallholder farmers that typically kept a proportion of their harvest for their own consumption and sold the remainder. |
| Recruitment | At each study landscape (square), we interviewed the farmers managing the fields upon which our four equally spaced points fell, as well as up to two more randomly selected farmers per landscape (see Supplementary Information 3). No farmers refused to be interviewed. We accounted for observable confounders by careful selection of study landscapes and statistical matching (see Methods and Supplementary Information 2). |
| Ethics oversight | Ethical approval was given by the Cambridge Psychology Research Ethics Committee (approval code PRE.2022.090) at the University of Cambridge prior to commencement. Before participating in the study, all farmers gave informed consent. |

Note that full information on the approval of the study protocol must also be provided in the manuscript.

# Field-specific reporting

Please select the one below that is the best fit for your research. If you are not sure, read the appropriate sections before making your selection.

☐ Life sciences  ☐ Behavioural & social sciences  ☒ Ecological, evolutionary & environmental sciences

For a reference copy of the document with all sections, see nature.com/documents/nr-reporting-summary-flat.pdf

# Ecological, evolutionary & environmental sciences study design

All studies must disclose on these points even when the disclosure is negative.

| | |
|---|---|
| Study description | We collected yield and farm management data from 'zero budget natural farming' (ZBNF) and agrichemical farmers, and conducted bird counts in these two systems as well as natural forests. We assessed the impact of ZBNF on yield, profit, and densities of different bird species, and we evaluated the associations between landscape-level yield, profit, and bird population outcomes for each farming system. |
| Research sample | In total, we worked in 51 landscapes (13 ZBNF landscapes, 13 agrichemical landscapes, 25 forest landscapes). We collected yield data from 206 harvests and profit data from 128 fields found within those landscapes, and we obtained density estimates of 199 bird species by conducting 816 point counts. We recorded all bird species present at our sites. Our samples are representative of ZBNF and agrichemical farming systems, and of natural forests in northern Andhra Pradesh. |
| Sampling strategy | We required large continuous areas to be exclusively farmed using ZBNF practices, and further site requirements to reduce the influence of possible confounders left us with 13 landscapes. We identified agrichemical and forest sites with similar biophysical and socioeconomic characteristics (see Supplementary Information 2 for details). |
| Data collection | We conducted 10-minute point counts and identified all birds present from visual and/or acoustic cues. We collected yield and profit data via a questionnaire with the farmers. See methods for details. |
| Timing and spatial scale | We conducted 128 structured interviews of farmers between December 2022 and February 2023. The interviews asked for detailed information about each identified field over a one-year recall period (beyond that recall issues may arise, see Supplementary Information 3). We conducted 10-minute point counts with no settling-in period at each point location in both the winter |

(December–March) and summer (April–June) seasons over two years (2021/2022 and 2022/2023). Hence, we visited each point on four separate occasions, totalling 816 repeats (51 squares * 4 points * 4 repeats) and 8,320 minutes observation (160 minutes per square) across all sites.

**Data exclusions**

Individual birds we only saw in flight and birds that flew in during the count period were excluded, and we removed observations greater than 100 metres from the point. We discarded records of bird species that point counts do not adequately sample, namely largely aerial and/or transient species (see Supplementary Information 4). We removed four tapioca harvests when examining ZBNF's impact on yield as they represented outliers. For the profit analysis, we removed seven fields that had estimated profits of over 25,000 INR per hectare as these were deemed implausible.

**Reproducibility**

No experiments were conducted.

**Randomization**

Farmers chose whether or not to adopt ZBNF prior to our study. We used statistical matching to control for observable confounders (see Methods).

**Blinding**

Farmers chose whether or not to adopt ZBNF prior to our study.

Did the study involve field work? ☒ Yes ☐ No

## Field work, collection and transport

**Field conditions**

Surveying birds in both the winter and summer seasons over two years allowed us to record species mainly vocal during the breeding season as well as winter migrants, and to control for seasonal and yearly variability. We avoided conditions of rain, fog, or high winds.

**Location**

Andhra Pradesh, India (districts: East Godavari, West Godavari, Vizianagaram, Visakhapatnam, and Srikakulam)

**Access & import/export**

Research was approved by ethics committees at the University of Cambridge (approval code PRE.2022.090). We did not work in strictly protected areas (IUCN category II).

**Disturbance**

Disturbance to birds and other taxa was minimised by using survey methods that are not invasive (i.e. purely observational).

# Reporting for specific materials, systems and methods

We require information from authors about some types of materials, experimental systems and methods used in many studies. Here, indicate whether each material, system or method listed is relevant to your study. If you are not sure if a list item applies to your research, read the appropriate section before selecting a response.

### Materials & experimental systems

| n/a | Involved in the study |
|-----|----------------------|
| ☒ | Antibodies |
| ☒ | Eukaryotic cell lines |
| ☒ | Palaeontology and archaeology |
| ☐ | ☒ Animals and other organisms |
| ☒ | Clinical data |
| ☒ | Dual use research of concern |
| ☒ | Plants |

### Methods

| n/a | Involved in the study |
|-----|----------------------|
| ☒ | ChIP-seq |
| ☒ | Flow cytometry |
| ☒ | MRI-based neuroimaging |

## Animals and other research organisms

Policy information about studies involving animals; ARRIVE guidelines recommended for reporting animal research, and Sex and Gender in Research

**Laboratory animals**

The study did not involve laboratory animals.

**Wild animals**

199 bird species were observed during point counts. See Supplementary Table 6 for a full species list.

**Reporting on sex**

No sex-based analyses were performed.

**Field-collected samples**

The study did not involve samples collected from the field.

**Ethics oversight**

No ethical approval was required for the bird surveys as these were entirely observational.

Note that full information on the approval of the study protocol must also be provided in the manuscript.

## Plants

Seed stocks

No plant material was collected.

Novel plant genotypes

Not applicable.

Authentication

Not applicable.

