## [Peer Review File · Nature Ecology & Evolution]

India's agroecology programme, 'Zero Budget Natural Farming', delivers biodiversity and economic benefits without lowering yields

Corresponding Author: Ms Iris Berger

Version 0:

Decision Letter:

20th January 2025

Dear Dr Berger,

Your manuscript entitled "Agroecological transition delivers win-win outcomes for people and nature" has now been seen by three reviewers, whose comments are attached. I apologise that it has taken longer than we usually aim for to return these reviews to you.

As you will see from the comments, the reviewers have raised a number of concerns which will need to be addressed before we can offer publication in Nature Ecology & Evolution. We will therefore need to see your responses to the criticisms raised and to some editorial concerns, along with a revised manuscript, before we can reach a final decision regarding publication.

The reviewers also commented, as part of their comments to editors, that they were not able to fully assess all of the analysis - in part because of their relative expertise(s) and in part because of additional information that they are requesting as part of the revision process. As a result, we will probably need to bring on an additional (fourth) reviewer when you submit your revised version. Our subsequent decision(s) will of course have to take into account the comments of that reviewer and their expertise. I'm sorry this will add an additional step, but given that the current reviewer panel are seeking substantial additional information and clarification regarding the methods, we felt that it would be more time efficient to ask for these revisions first, before we seek further input from them and an additional reviewer.

We therefore invite you to revise your manuscript taking into account all reviewer and editor comments. Please highlight all changes in the manuscript text file.

* If you have not done so already please begin to revise your manuscript so that it conforms to our Article format instructions at <http://www.nature.com/natecolevol/info/final-submission>. Refer also to any guidelines provided in this letter.

* Extended Data Figures - please ensure that any supplementary figures and tables that are crucial to the manuscript's conclusions are converted into Extended Data figures and tables to increase visibility of these data. Extended Data figures and tables are online-only (present in the online PDF and full-text HTML versions of the paper), peer-reviewed display items that provide essential background to the article but are not included in the main article due to space constraints. A maximum of ten Extended Data display items (figures and tables) is permitted.

Link Redacted

Nature Ecology & Evolution is committed to improving transparency in authorship. As part of our efforts in this direction, we are now requesting that all authors identified as 'corresponding author' on published papers create and link their Open Researcher and Contributor Identifier (ORCID) with their account on the Manuscript Tracking System (MTS), prior to acceptance. ORCID helps the scientific community achieve unambiguous attribution of all scholarly contributions. You can create and link your ORCID from the home page of the MTS by clicking on 'Modify my Springer Nature account'. For more information please visit www.springernature.com/orcid.

[redacted]

Reviewer expertise:

Reviewer #1: agroecology, sustainability

Reviewer #2: biodiversity conservation, land-use

Reviewer #3: agroecology, ecosystem services

Reviewer comments:

Reviewer #1 (Remarks to the Author):

This paper evaluates the effects of an agroecological program called ZBNF, zero-budget natural farming, that is being rolled out at large scale in India, on farming yields, profits and bird biodiversity, in order to provide information on the benefits versus trade-offs of this farming method between food production and biodiversity. As agriculture is the most impactful single human activity affecting biodiversity, but is necessary for human survival, finding methods that are least harmful for biodiversity while maintaining yields and profitability are critically needed. Further, the appropriate farming methods and scale of farming versus protection has been a main debate in the conservation literature for over a decade (land sparing/land sharing debate). This important paper sheds light on this debate and provides a rigorous sampling and analytical strategy for assessing tradeoffs and synergies between biodiversity and agricultural production. I have not seen other papers examining this tradeoff that have implemented this degree of rigor in the data collection and analysis.

The paper achieves a methodological high bar in its site selection, field methods and analytical techniques. For example, the authors used methods of causal inference (matching) from econometrics, coupled with a rigorous site selection process, to control for confounding variables between the ZBNF and the agrichemical farming landscapes' effects on bird biodiversity, farmer yields and profitability. They further estimated species densities using distance techniques and Bayesian hierarchical modeling (as opposed to simply measuring occupancy or abundance as many bird studies do), and they integrated ecological and socio-economic data collection and analyses successfully.

A main conclusion from their work is that agroecological methods (specifically, ZBNF) do not penalize yield but improve profits (and thus presumably could improve livelihoods) while improving conditions for farmland birds relative to agrichemical farms. The improvement of profits in the absence of yield increases is surely due to the use of locally produced inoculants instead of purchased agrichemical inputs. Another important conclusion is that many bird species do not inhabit farmlands, and thus efforts to promote ZBNF must be coupled with policies to protect existing forests, so that forest encroachment is not exacerbated.

Revisions are needed in the description of the methods and results as they are often explained too briefly in the main text, leading to confusion. Some details are provided only in the supplement that are necessary for comprehension of the paper in the main text.

Detailed Comments

Some additional details are needed in main text.

1. L. 49 following. Does ZBNF only change the use of inoculants versus agrichemicals? What other agroecological techniques are commonly used along with ZBNF (even if it does not specify that these must be used). I see these are described in the supplement. (especially those that are encouraged to be used, but not part of the main method)
2. L. 74 Briefly define g-computation
3. L. 75. What do you mean by yield and economic profit measured respectively at 'harvest level' and 'field level' -- briefly describe.

Additional comments

Why would ZBNF increase frugivore abundance? Are different crop species including fruit trees being planted?

Related to the above, did you also ask ZBNF farmers and/or note during ground-truthing which of the additional techniques described in the supplement (besides the 4 pillars) were being used? In particular live fences, added trees, etc would be most likely to affect bird populations. If so, can you provide this data in the supplement? Could it be incorporated as a co-variate into analyses (perhaps just within the ZBNF farms, since presumably these methods are not in use in the agrichemical landscapes)? These may be important for explaining the increase in frugivore abundance, along with the proportion of tree-crops.

I. 41 This statement is incorrect. There are many agroecological studies that have been conducted on real farms. A compilation of data from real farms using a spectrum of agroecological techniques was recently published, for example, in Rasmussen et al. 2024 in Science.

Fig 3A. define the grey color (opaque was unclear)

I. 235. Here you can cite prior work emphasizing the need for agroecological techniques and ecological intensification approaches to be coupled with policies for protecting natural habitats in order to best protect biodiversity e.g. Kremen 2015 Reframing the land-sparing/land-sharing debate for biodiversity conservation. *Annals of the New York Academy of Sciences*, 1355:52-76. DOI:10.1111/nyas.12845.

I. 504 -508. I think that you mean that after groundtruthing 28 potential ZBNF sites you excluded 15, leaving 13. However, the exclusion criteria at this point in site selection are not described and the sentence itself was not clear.

I. 540 main text mentions subsistence farmers as being different from agrichemical farms but but in methods no mention of subsistence farms are made, please clarify here (and not in the supplement alone)

I. 542/543. Clarify whether a harvest refers to everything harvested from a plot of land over time, or whether each harvest (of successive crops) is considered and included separately, and if so how. Clarify also how you can bring harvest of different crops together in the same analysis of yields? [This is noted in supplement but should also be briefly mentioned in main text]

I. 591 this line is not clear to me. We matched and conducted post matched analysis. I think first you should just describe the matching procedure, and then later, describe the analyses.

I. 610. The sequence of matching is not really clear. I think that including a flow diagram would help. I understood that you selected 13 ZBNF squares based on a set of criteria (that still need fuller explanation) and then matched those squares with 13 agri-chemical squares (presumably matching on same criteria). Yet this section seems to imply that you matched on yield and on economic profit, which wouldn't make sense since those are your response variables.

What is also confusing is that you then say in line 610, "prior to matching"...so this makes me believe that I do not understand your matching procedure. You also say that your samples were relatively similar already because of the site selection process.

Finally, it is not clear to me whether 'matching' means that each ZBNF field is matched with one agrichemical field or whether the pool of ZBNF fields is matched with the pool of agrichemical fields.

As ecologists are less familiar with matching procedures it would be helpful to clarify this section in main text.

L 632 following. Brief explanations of the distance and genetic distance methods would be helpful

Supplement

L 97 the important point about the 5 km surrounding landscape also having 75% ZBNF (for ZBNF farms) should be moved into main document methods

L 380-384. Some of this information would have helped me to better understand the procedure used

It would be helpful to have a list of crops and their prevalence in each sample type as a table

Figure S2 and Figure S3 legends should distinguish to which analysis they respectively apply (e.g. Yield, Profit).

Reviewer #2 (Remarks to the Author):

Overall, I think this is an impressive body of work describing the impacts of transitioning to ZBNF practices on agricultural productivity, economic profits of farmers, and bird biodiversity. The authors clearly spent a lot of time on design, field sampling, analyses, and interpretation of results. These types of studies are rare, linking alternative farming practices to both socioeconomic outcomes and biodiversity - but they are incredibly important, as this bigger picture is so often missing. While I think that an updated version of this manuscript could be suitable for publication in *Nature Eco Evo*, I have many issues with the manuscript and analyses in their current state.

I really like this manuscript and this study, and think it is important to publish. As such, I was very thorough in my review of this manuscript, hence the extensive critiques I provide below.

More Concrete Examples of Species- or Trait-Level Mechanisms

Throughout the Results/Discussion, the language is overly focused on statistical results rather than the biological interpretation of these results. For example, in lines ~355–380 (Results on farmland birds), the authors mention broad guild responses but provide few examples. For instance, they say 17 species responded positively, 6 negatively, 91 “uncertain.” A deeper look could reveal, for instance, that “Species X, a ground-nesting insectivore, benefited from greater invertebrate prey in ZBNF fields lacking pesticide treatments.” This tangible example would help in explaining why certain birds respond differently. Additional trait-based analysis (e.g., from AVONET’s morphological or dietary data) would help show if larger-bodied omnivores, waterbirds, or certain insectivores (eg Sallying or gleaning) are the main winners vs losers. Further analysis will be important for pulling mechanistic insights from these trends, as this additional information tells us a lot more about the ecology of the species. I discuss some of this throughout the sections below,

Forest vs ZBNF Conservation Value

The discussion hints that forest specialists still need intact forest. But the authors don’t dissect how “forest condition,” “farm structure,” or “landscape context” might interplay. Deeper insights into how farmland structure (e.g., tree cover, hedgerows) matters relative to chemical inputs. The section “Forests hold irreplaceable conservation value” (around lines ~352–383 in the main text) is short. They do not detail whether these forests were mature old-growth or secondary. Nor is there mention of micro-habitats (e.g., wetlands, snags). Mechanistically, structural complexity on farms (e.g., boundary vegetation, hedgerows, scattered fruit trees) are key for bird diversity and abundance. The authors never measure or discuss these habitat features, omitting an important aspect of farmland ecology.

Sampling Design and Field Methods

The authors state they visited each point four times over two years, covering winter and summer. Bird communities can vary year-to-year due to local migration, rainfall patterns, and habitat changes. A two-year snapshot, with no baseline (“pre-ZBNF”) data, makes it difficult to assert causality or trend stability. Any unaccounted interannual variation (e.g., an unusually wet or dry year) might confound apparent differences between ZBNF and conventional sites.

The authors also surveyed 25 “forest squares” to gauge how far either farming system deviates from “natural” conditions. However, “forest” is treated monolithically. Forest quality, successional stage, and disturbance history are not detailed. Heavily logged secondary forests vs. mature continuous forest can differ drastically in avian assemblages. Without matching between farmland and forest squares (they only mention matching farmland squares with each other?), there could be hidden site-level or landscape-level confounders (e.g., forests on steep slopes or at higher elevations) that influence bird community differences.

Methods

Complexity of Distance Modelling and Low Samples Sizes

The authors note a “two-stage” approach, fitting species- or guild-specific detection functions and then modeling abundance (or density) based on truncated distance bins. In principle, distance sampling is a strong method. In practice, it is sensitive to sample sizes, observer skill, accurate distance estimation, and consistent habitat structure. Many farmland birds are small and cryptic, or predominantly detected by calls. The paper does not fully describe how they handled aural vs. visual detections in distance estimates, or how they confirmed distances for fleeting or fast-moving species.

For rarely encountered species (< 40 observations), the authors group species based on certain traits and fit a common detection function. If grouped species differ significantly in flightiness, detectability, or typical habitat strata, this approach can bias densities upward or downward for particular species. Adjusting detection functions by habitat (forest vs. farmland) may help, but farmland itself can span a gradient from open fields to interspersed trees, which the authors do not detail. Varying detectability within farmland habitats might go unaccounted for.

The authors also tested half-normal vs. hazard-rate key functions with polynomial or cosine adjustments, using AIC to select the “best fit.” This can be a valid approach, but small sample sizes per species (or per group) can lead to unstable parameter estimates, especially if many species are uncommon. In addition, overfitting or inconsistent model selection across species can lead to spurious density estimates if not cross-validated. The paper does not mention cross-validation or goodness-of-fit diagnostics (beyond AIC). The authors should include these in the updated manuscript. Given that all downstream bird diversity/abundance analyses are based on these density estimates, it is important to check the quality of these estimates.

Many models, minimal explanation

The authors run a hierarchical zero-inflated Poisson model (ZIP) for trophic guilds, a zero-inflated Poisson for species-level analysis, additional models for yield/density trade-offs, plus comparisons to forest. There is a mention of “five different Bayesian models” (Methods lines ~310–320). It can be overwhelming and difficult to follow which question each model is answering.

The manuscript lumps multiple advanced analyses together: matching, zero-inflation, hierarchical random slopes, etc. The rationale for each step is scattered across text and supplements.

Further, the authors do not clarify how seasonal data (winter vs. summer) feed into these models—e.g., Are they pooling data across seasons or specifying season as a random effect? (Seasonality is crucial in tropical agricultural landscapes!!)

I would suggest the authors present a table or schematic clarifying each model and what it addresses/what hypothesis it tests. In addition, the authors must explicitly state how seasonal differences in bird presence/behavior were incorporated into the structure of the models.

Removal of “Extreme”/“Unrepresentative” Data

The authors mention removing tapioca harvest outliers and “extreme” values. The authors do not specify what threshold or biological rationale guided these exclusions.

Eliminating outliers can bias the results if “extreme” yields/profits reflect real phenomena (e.g., very high-yield conventional rice fields). Potentially, this might artificially narrow the difference between ZBNF and conventional. Transparency about the distribution of outliers, the numeric cutoff, and the re-analysis with/without outliers would help.

Forest Site Selection and Clustering

The authors provide little detail on how forest squares were chosen, yet the type/quality of forest strongly affects bird communities. Visual inspection of Supplementary Figure 1 suggests clustering of these forest sites and multiple forest types. If forest squares differ widely (e.g., some are moist deciduous, others dry), or if they are in different elevations, the resulting “forest bird community” might not be comparable. Some sites might be near farmland edges, others deep inside protected areas. This can inflate or obscure differences relative to farmland squares. No mention of controlling for forest type in the matching process or in the modeling.

Trait data and guild usage

The authors grouped species into functional groups for detection modeling and for some of the main results (granivore, frugivore, invertivore, vertivore, omnivore). I suspect important species-level variation is being lost by this grouping. For example, vertivores presumably are carnivorous birds that eat small vertebrates (raptors, large kingfishers, etc.). They may have drastically different foraging behaviors, territory sizes, detectability, or habitat requirements. Lumping them might be too coarse. The entire analysis of guild-level responses might mask crucial distinctions (e.g., small insectivores in field margins vs. canopy-gleaning insectivores around orchard trees). The authors should consider additional analyses, such as a multi-trait approach that might better explain observed responses. In addition, for the detection modeling specifically, show a short table of which species were grouped, why those species share detectability traits, and how final detection functions performed.

Seasonality

As mentioned above, farmland structure, phenology, and bird presence vary seasonally, especially in the tropics. The authors note they surveyed “winter and summer” but do not clearly explain how this was modeled. If, for instance, ZBNF fields have different crop rotations or vegetation cover in the winter vs. summer, bird detection or even presence/absence might differ strongly. Lumping these data could blur seasonal patterns. Without a seasonal random effect or a seasonal interaction with farming system, one might conflate differences that are purely seasonal with “treatment” effects. As a starting point, it would be helpful to know how farmland bird communities shift from winter to summer in each farming system. However, controlling for phenology in the modelling approach seems warranted as well.

Overarching Concerns about Model Complexity and Justification

The authors use multiple advanced models without a lot of clear explanation of why and what it tells us.

ZIP Structures:

The authors use zero-inflated Poisson models to account for excess zeros. ZIP can be valuable when a substantial fraction of point counts return no detections of certain species. However, zero-inflation can arise for many reasons—true absence, detectability failures, or extremely patchy distributions. It may also be the result of a missing covariate. By combining all farmland points in a single hierarchical ZIP model, the authors assume a consistent zero-inflation process. If agricultural

vs. ZBNF squares differ systematically in vegetation or detectability, the zero-inflation component might conflate real biological absence with reduced detectability in certain habitats.

Multispecies, Multiguild Structure

The authors nest species within trophic guilds (granivore, frugivore, omnivore, invertivore, vertivore), and then treat farming practice as a fixed or varying slope in the Poisson portion. While this can produce an “average guild-level response,” it can mask intra-guild variation. Some insectivores thrive in farmland edges, others require more tree cover, etc. Lumping them might smooth out truly different responses and produce overly simplistic “guild” effect sizes.

The authors also attempt to interpret species-specific slopes from random effects. But with many species (114 farmland species) and relatively few replicates, the random-slope estimates can be noisy or heavily shrunken toward the overall mean, reducing confidence in species-level inference.

Matching for the Avian Analysis - habitat structure on farms

For the farmland bird comparison, the authors mention matching each point location on elevation, temperature, and native vegetation. However, farmland structure (e.g., presence of hedgerows, orchard trees, farm ponds) may be key for bird communities. If these features differ systematically between ZBNF and conventional sites but were not used as matching covariates, the results attributing bird differences to “ZBNF” might be confounded by local habitat variations.

The forest points are not matched at all, only farmland squares are. So comparisons of farmland vs. forest bird communities are purely observational and could be influenced by local geography or other unmeasured factors.

The paper ascribes bird abundance increases in ZBNF primarily to reduced pesticide use or higher resource availability (insects, fruits, etc.). However, the authors did not measure pesticides, insect abundances, or vegetation composition. Therefore, the ZIP results remain correlational. Distinguishing which specific management practices—reduced pesticide spraying, diversified cropping, presence of trees—drive these patterns is speculative.

Figures 2: Visualization and Interpretation

In Figure 2a–b, the shading to show “strength” of trend duplicates information already reflected in the 95% BCIs. This visually overcomplicates the results without adding new information. A more intuitive scheme could color-code guild or habitat affinity (forest specialists, grassland/wetland species, farmland generalists). rather than “95% PD share.” This would convey new insights rather than re-stating statistical certainty.

Figure 3: Trade-offs with yield

The authors claim there is “no negative effect of yield on density in ZBNF.” Yet the average trend lines (for all but one guild) are negative, just not statistically significant. In Figure 3a (and Extended Data Figure 1b), the lines for granivores, invertivores, and omnivores in ZBNF show negative slopes, albeit wide BCIs. “Not significant” does not equal “no effect.” The narrower intervals in conventional farms vs. wide intervals in ZBNF might reflect either truly higher variability in ZBNF sites or the model insufficiently capturing the underlying heterogeneity across farms. Simply concluding “no negative effect in ZBNF” oversimplifies the result. The ecological interpretation should be along the lines of: “While the negative slope is weaker or more variable under ZBNF, trade-offs are not entirely absent.”

I would suggest that the authors revisit the language in the main text (e.g., lines describing Figure 3 or lines ~398–417) to not overstate a “lack of trade-off.”

Claims about “synergies” or “neutralizing trade-offs” with yield should be tempered by acknowledging that short-term partial data on a single taxonomic group (birds) may not reflect broader biodiversity or future yield stability.

Uncertain: In the manuscript and in Figures 2b and Extended Data Figures 1, 2, “uncertain” appears to label species whose Bayesian credible intervals overlap zero (i.e., no strong statistical support for positive or negative trends). However, “uncertain” is not standard terminology in ecology or statistics.

Replace “uncertain” with clearer language, e.g., “no detectable difference relative to 0,” or “no significant change.” It would also be great to get more explicit biological interpretation: do these “neutral” species share certain traits or habitats?

Reviewer #3 (Remarks to the Author):

General comment to authors:

Overall, the paper is interesting and explores an area worth investigation: how to carry out high-yield farming without limited impacts on biodiversity. Despite some major reservations, the authors should be able to address them to make the manuscript worth publishing. My major concerns go to i) the overly positive tone used, ii) the lack of information in the main manuscript (i.e. not in supplementary material) on the methodological sampling setup and the ZBNF programme.

With regard to the tone used, authors tend to over-interpret their results sometimes verging on propaganda. For instance, presenting the ZBNF programme as delivering win-win outcomes for biodiversity and people, while results show an absence of significant negative relationships between biodiversity and profit/yield is to me an over-interpretation as this is different from a positive relationship. Though non-significant, the relationships remain mainly negative. Despite encouraging

results with regard to the ZBNF programme, which indeed removed the negative relationships, they do not show any 'win-win outcomes', it's rather a 'win-less-loss' outcome.

A lack of clarification is critical with regards to the methodological setup. A short but complete description of the sampling setup should appear from the outset of the paper to help readers understand the result section. If not required by the journal, I would suggest moving the method section before the result one. If that is against the journal's requirement, please, still make sure readers have a clear view on the sampling design and what the ZBNF programme entails before diving into results. As for the method section, it could on many part be shorten for the sake of clarity (see detailed comments for suggestions).

With regards to the ZBNF programme, authors should be more specific about the measures supported through the programme and how supported farms distinguish from 'agrichemical' ones. As authors refer to the BAU farm model as 'agrichemical', one could be tempted to deduct that the difference lies mainly and only in the use or non-use of chemicals for pesticides. However, when reading further about the ZBNF programme (supplementary material 1), we discover that practices entailed by the programme also include reducing soil tillage intensity, and mulches, thus going beyond a mere substitution of a chemical product with microbial inoculums. The ZBNF programme and agricultural practices covered by it should be introduced at the outset of the paper. The programme aims at reducing pesticides through four main axes, but authors fail to report whether each farmer engaged has been applying these axes in the same way and for the same amount of time. It is also unclear if the practices applied allow to get rid of 100% of pesticides or whether they just allow a decrease in dependency upon them.

In the same line of thoughts, authors tend to present pesticides as the only driver of biodiversity loss in agriculture, ignoring other major drivers: mechanization, landscape homogenization, to only cite a few.

In general, I question the choice of birds as bio-indicators as they are able to travel long distances, I wonder how they could be representative of 500m² authors are comparing. As practices introduced through the ZBNF programme mainly relate to soil, a soil-dependent organism would have seemed more appropriate.

Additionally, I question the relevance to include forests in the comparison. Agricultural systems are by essence modified by humans and no matter the level of 'agroecologization' they will always be more impacted ecosystems than natural ones such as forests. I understand the temptation to include such comparison which provide very contrasted results, but am not convinced of the intellectual added value.

Detailed comments

L9: how sure can you be that this is the largest agroecological programme worldwide? Is it 'largest' in terms of area covered? Amount of finance / subsidies distributed ? Amount for farms supported? Please, provide references and precisions and/or nuance.

L34: + mechanization, landscape homogeneisation and simplification.

L41: please nuance. 'a large share' would be more appropriate, but you could also acknowledge that a growing body of research is endorsing systemic approaches.

L48: again nuance 'world's largest' or provide reference

L52: More information on the practices applied in the ZBN programme would be welcome in main text. I would include the enumeration of the 'four wheels' listed in Supp 1. At this stage, you only mention bio-inoculant and make no reference to mulching and reduced tillage practices.

Also, the time since the programme has been put in place should be reported.

L54: what is meant by 'incentivised'? To they give money? Training?

L58: birds should be introduced earlier as why they would be an interesting group to study.

L70: what is 'careful site selection'? What are your selection criteria?

L73: please clarify the share of ZBNF and agrichemical within your '206 harvests and 128 fields'. Please also clarify whether all those fields beyond to different farms. Please also specify which crops are being covered for each system type.

We need some extra explanation before diving into results.

L87-89: this sentence should be clarified.

L90: the added value of mentioning where the 2.6% highest yield belonged to is questionable

L100: 'perverse subsidies' are mentioned several times throughout the paper, please clarify what is meant by this.

L103: these are central pillars of sustainable food production systems in general, in my sense, there is no need to specify 'for marginalized smallholders in the tropics' (see the 13 principles of Agroecology underpinned by the HLPE of the FAO: <https://openknowledge.fao.org/server/api/core/bitstreams/ff385e60-0693-40fe-9a6b-79bbef05202c/content>)

L 106: it is unclear what is meant 'across heterogenous conditions', you refer right after to 'individual ZBNF interventions', do you rather mean 'across the heterogeneity of if practices covered by the ZBNF programme'?

L111: same number of crops, what about crop types? Do they grow the same crop species between ZBNF and agrichemicals? Also, same amount of crops could hide intercropping methods (i.e.

114: what is a 'yield stabilizing effect'?

L117: for this section about birds, since these can fly long distances in a short time frame, it should be clarified how ZBNF parcels are arranged in space. Are they gathered together, offering a fairly large (how large?) plot of parcels under ZBNF or are ZBNF parcels scattered across the landscape within an agrichemical matrix?

L119: temporal or geographical repeats ? Do the 104 count include this 4x replication?

L119: what about forests ecosystems?

L119: how does a "point" covers hectares?

L126: unclear what is meant by 'functional traits governing their persistence in agricultural landscapes'

L128 & 154: at this stage, 'habitat preference' is only treated in supplementary material, it should be treated in main

manuscript as well (or drop it completely).

L163-164: you mention both the level of the farming system and the landscape level, which are two different scales, please clarify.

L170: what is 'an uncertain response'? Unsystematic maybe?

L170: 'tradeoffs' instead of 'declines'?

L175 and L183: you mention 'landscape-level' yield and economic profit assessment, while in L75 you mention harvest level yield and field-level economic profit assessment, are these different estimates or are landscape-level estimates extrapolations of the harvest/field ones?

L183: I would double check results for vertivores which is oddly opposite to all other guilds.

L185: 'responded' sounds like it was a causal relationship, while it's not necessarily.

L197: I would not use the term "synergies" since birds seem to remain negatively affected

L202: you have '25' and 'twenty five' standing together

L202: add 's' to hectare

L210: what is community integrity?

L216: replace 'shifts' with 'declines'?

L388: replace bar-chart by boxplot

L408: specify color code

L409: wouldn't it be useful to have sp names directly in the figure?

L412: Figure 3: please check data for vertivores vs yield, results seem odd. If right, then please provide with some hypotheses of why such contrasted results for this group.

L418: add 'significantly' before 'decreased'

L473: general comment on methods: it feels like some parts could be shorten for the sake of clarity. Sometimes, 'less is more'; too much information can blur the message.

L486: what is 'careful site selection'

L496: authors should refer to this scale of analysis: 'ZBNF villages with at least 80% farmers involved in the programme' earlier in the paper, otherwise the readers think you compared isolated ZBNF parcels in a agricultural landscape.

L505: what is meant by 'semi randomised'

L506: on which range is calculated this natural vegetation cover? The 500m² plots?

L512: these are very small parcels, are the crops grown in ZBNF similar to agricultural ones?

L539: in L511 you mention that each square on average contains 174 fields, which leads to 26(squares)x174= 4524 fields in total approximately. How do you explain you only gathered information on 128 fields? How are these representative of the 4396 others?

L550: these are pseudoreplicates, not replications

L553: contribution of authors to the different tasks should probably not be mentioned as part of the main text body, but can be kept to the 'authors' contribution' section.

L576: replace 'behavioral characteristics' by 'habitat preference'.

L601: what is 'agroecological suitability'?

L610-615: how do you justify you remove outliers? Outliers are data, which should be treated as all other data, they can only be removed if you can be sure there are the results of sampling / measurement mistake because their results are implausible.

Supplementary material

L19: the fourth wheel '(4) 'whapahasa' – improving soil structure and reducing tillage through soil aeration' isn't clear. Isn't the other way around: your increase soil aeration by reducing soil tillage?

*****END*****

Version 1:

Decision Letter:

7th July 2025

Dear Iris and co-authors,

Thank you again for your patience while we gathered the reviews on your revised manuscript currently entitled "Agroecological transition delivers win-win outcomes for people and nature" (NATECOLEVOL-24102918A). It has now been seen again by the original three reviewers and a new reviewer; their comments are below. The original reviewers find that the paper has improved in revision, and the new reviewer has only relatively minor suggestions. We are therefore happy in principle to publish it in Nature Ecology & Evolution, pending minor revisions to satisfy the reviewers' remaining suggestions and to comply with our editorial and formatting guidelines.

We expect to be able to assess your responses to the new Reviewer 4 ourselves, without needing to seek further advice from that reviewer, but we will go back to them if we have any questions.

You're welcome to start a 'response to reviewers' document at this stage, and begin incorporating your responses to the reviewers in your manuscript, but please don't resubmit anything until you have received our final editorial requests in the form of a checklist that we'll get to you within around one week.

If you have not done so already, please ensure that you also email us a completed copy of the Reporting summary :

Reporting summary: https://www.nature.com/documents/nr-reporting-summary.pdf

We will perform detailed checks on your paper and will send you a checklist detailing our editorial and formatting requirements in about a week. Please do not upload the final materials and make any revisions until you receive this additional information from us.

[redacted]

Reviewer #1 (Remarks to the Author):

The authors have done a great job responding to my critiques by clarifying a number of methodological points and points about the farming system itself. They also added much more information on the natural history of the birds in response to another reviewer. The paper looks to be in good shape.

I just have one question -- on Supplemental Fig 1, the y-axis is number of harvest per year. The numbers go as high as 60. This doesn't seem possible if this is number of harvests per field per year. Is it perhaps at some larger scale? The scale needs to be stated.

Reviewer #2 (Remarks to the Author):

I thank the authors for thoroughly addressing my previous comments. They have significantly improved clarity, provided robust methodological clarifications, performed additional analyses (including trait-based explorations and seasonal robustness checks), and refined their interpretations. The revisions to the discussion of trade-offs are well done, clearly distinguishing between mitigation rather than elimination of trade-offs, and avoiding any undue overstatement of synergies. The manuscript now carefully balances ecological insights with appropriate statistical caution.

There remain only a few minor issues to clarify or correct before the manuscript is publication-ready. Firstly, please ensure consistency in terminology when describing species responses. The manuscript occasionally still uses the term "uncertain" to describe non-significant responses (e.g., line 197 in the Results). Please revise all occurrences of "uncertain" to "non-significant," or a similar phrase, to clearly communicate the statistical outcome.

Second, in the Results section, lines 169 and 179 use the word "compromising" instead of "comprising" when describing the composition of bird guilds.

"Insessorial" may not be immediately clear to all readers. A brief clarification or substitution with a more common synonym such as "perching (arboreal)" would improve the readability.

Apart from these minor editorial clarifications and corrections, the manuscript is very well done. Once these points are addressed, I fully support acceptance of this manuscript to Nature Ecology & Evolution.

Thank you

Reviewer #3 (Remarks to the Author):

General comment: I appreciate authors have adequately addressed all my comments and recommendations. The manuscript now reads very fluently and clarity has clearly improved on the experimental setup and methodological approach. The paper is interesting and very well presented.

Disclaimer: I accept no responsibility for the validity of statistical operations that fall outside my experience and expertise. I

would personally have performed multivariate analyses as usually done in ecology and communities analyses. Yet, these are presented in a robust and logical way.

My remaining comments are only minor:

L83: you refer to questions, while you present objectives on L69: please align wording.

L120: should 'produce' be replaced by 'products'? (to be checks with English native speaker)

L121: not clear what is meant by 'reflecting the state-wide situation. The phrase could do without that vague piece of information

L122: replace 'perverse' by 'agricultural' subsidies to be coherent with previous wording and appear less subjective.

L244: the fact that you have checked for landscape heterogeneity and that you can therefore conclude that density-productivity curves are largely driven by differences in field-level management is a big result you should emphasize more. This piece of information should be added to your abstract. A large body of literature has focused on landscape influence on biodiversity, your study is novel in the way it proves that the agricultural matrices is also very much important in biodiversity conservation.

L744: this paragraph belongs to results not method

Reviewer #4 (Remarks to the Author):

Review

This is a very thorough, well-developed study, and the counterfactual methods have been very well implemented. I commend the authors on their rigour. There are however a few issues/questions that would benefit from clarity. This will make the paper stronger, and a point of reference for researchers interested in implementing counterfactual methods in agroecological studies.

Please note that I am mostly providing feedback on methods, and socio-economic (profit) estimations, as these are my area of expertise. My comments are as follows:

1. The authors calculated income using the ten-year mean wholesale state-level prices per unit weight of each crop between 2013-2023. This is an interesting choice, which needs further justification and explanation. Why choose 10 years when the intervention was implemented in 2017? I suspect that averaging over 10 years was chosen to dampen price fluctuations, but a more reasonable choice would be to calculate income since that date (~5 years). Otherwise, your profit calculations are based on a variable that was measured prior to intervention. This might be justified if the fluctuations are consistent year on year, or if the 10-year average is the same as the 5-year average – or if the 10-year mean is the only data available. So please do justify this choice clearly.

2. To estimate the effect of ZBNF on profit, the authors only match on agricultural suitability, and justify this (in Supplementary Information 5) by explaining how some confounders are captured/controlled for by how they calculate profit. However, there remain a number of additional confounders that they do not include in the matching (for profit estimation). Because profit is a function of yield (as well as prices and costs), it follows that the same confounders that are expected to affect yield, should also be expected to affect profit. The authors report conducting additional matching with the other covariates, as a robustness test, and refers the reader to Supplementary Figure 9 (which shows results of balance tests). If the conclusions about impact are the same, then the reader must be able to verify this. Ideally, the paper would present the impact estimates that were produced using the full set of covariates used in matching. However, recognising this may involve lots of marginal changes that amount to no major change in conclusions – then at the very least (assuming the estimates are statistically equivalent) the authors should add a figure/table showing the profit estimates with and without the full set of covariates used in matching in Supplementary Information 5.

3. Supplementary Table S2 is good because it provides details about why the confounders is expected to influence treatment assignment (or in this case, adoption) and outcomes. This is critical information that many papers simply skip over, even though controlling for confounding is the main objective of counterfactual methods. However, because this is such an important issue, I would ask the authors to go a bit further, and clarify for each confounder exactly the expected direction of influence for the both treatment assignment and outcomes. I could then use this as an example to my team about how to report confounders.

4. Line 233 of the Supplementary Information: you exclude tenure costs. Please explain and justify this choice - do most farmers own or rent? If land rental is high, and/or varies by area, then this may explain/affect results.

Minor comments:

Line 88: replace "cannot quantify" to "do not account for". Quantifying system-wide effects is difficult with counterfactual methods anyhow. What these methods do, if implemented correctly, is control for these effects.

Line 138 – do you have a figure (proportion) of agricultural farmers that use cover crops? I'm assuming none do, but it would be helpful to confirm this.
